# Cysteine-enabled cleavability to advance cross-linking mass spectrometry for global analysis of endogenous protein-protein interactions

Fenglong Jiao[1], Merav Braitbard[2], Clinton Yu [1], Ben Shor[2], Bjorn-Erik Wulff[3], Dina Schneidman-Duhovny[2] ✉ & Lan Huang [1] ✉

Cross-linking mass spectrometry (XL-MS) is a powerful technology for probing protein-protein interactions (PPIs) and elucidating architectures of protein complexes at the systems level. While successful, the proteome coverage remains limited. To expand the scope of global PPI profiling, we introduce an innovative cysteine-based cleavable XL-MS platform using non-cleavable heterobifunctional lysine-cysteine (K-C) cross-linkers. The oxidation-induced transformation of cysteine cleavability enables unambiguous identification of cross-linked peptides. This strategy has been successfully applied to proteome-wide XL-MS analysis of intact cells, and its broad applicability has been demonstrated using three heterobifunctional cross-linkers. A total of 25,401 unique linkages from 2007 proteins have been identified, significantly expanding the existing XL-PPI map and increasing the interconnectivity of the human interactome. The XL-data generated here has been coupled with AlphaFold-based predictions and integrative modeling to reveal the structural characteristics of cellular networks, offering insights into the organization of native protein complexes in cells including the SERBP1-ribosome and eEF1A1-eEF1B complexes. Due to the effectiveness in modulating cysteine cleavability, our work presents an avenue for developing bifunctional/multifunctional cross-linking reagents to further advance XL-MS technologies. Additionally, the same strategy can be easily adapted to facilitate the characterization of cysteine modifications, reactivity and interactions to benefit chemical proteomics.

Cross-linking mass spectrometry (XL-MS) is a powerful and versatile technology that has proven effective for mapping PPIs in vitro and in vivo[1–6]. In comparison to other methods, XL-MS is unique in its ability to directly capture interactions from their native environments and derive endogenous PPI networks at the systems level with high throughput without cell engineering. Identification of cross-linked peptides permits simultaneous determination of both PPI identities and their interaction contacts, thus uncovering direct interactors and yielding protein structural details at residue-level resolution[1,3,5]. In addition, cross-links provide distance restraints to facilitate integrative structural analysis of large protein complexes by refining existing structures and/or assisting de novo structural modeling[2,5]. Deep

[1]Department of Physiology and Biophysics, University of California, Irvine, CA, USA. [2]The Rachel and Selim Benin School of Computer Science and Engineering, The Hebrew University of Jerusalem, Jerusalem, Israel. [3]Department of Biochemistry, Stanford University, Stanford, CA, USA. ✉e-mail: dina.schneidman@mail.huji.ac.il; lanhuang@uci.edu

learning-based structure prediction tools such as AlphaFold2 (AF2) have been coupled with XL-MS to facilitate detailed characterization of cellular networks, revealing molecular mechanisms underpinning protein structure and function in cells[7–10].

To enable successful XL-MS analysis, a diverse array of cross-linkers has been developed to target specific and/or non-specific amino acids[3,4,11,12]. Among them, homobifuctional cross-linkers have been extensively used for global XL-MS studies. Although multi-chemistry cross-linking has proven beneficial for expanding PPI coverage, the potential of heterobifunctional cross-linkers for proteome-wide XL-MS analysis remains largely unexplored. Given the effectiveness of lysine- and cysteine-targeting homobifunctional cross-linkers in global PPI profiling[13–18], it is anticipated that lysine-to-cysteine (K-C) heterobifunctional cross-linkers would uncover additional interaction interfaces to further increase PPI coverage. Therefore, developing new heterobifunctional K-C cross-linkers would be advantageous for advancing PPI studies, allowing for a more detailed description of protein interaction landscapes in cells.

While various bioinformatics tools have been developed to facilitate the identification of non-cleavable cross-linked peptides, MS-cleavability is critical for proteome-wide XL-MS analysis due to its ability to facilitate cross-link identification in complex samples[3,19,20]. Thus, the development of MS-cleavable cross-linkers has significantly advanced XL-MS studies, especially for global PPI profiling in vivo[17,21–24] and in vitro[15,18,19,25,26]. Previously, we have developed a suite of sulfoxide-containing, MS-cleavable homobifunctional cross-linkers targeting lysines[21,27–29], acidic residues[30] and cysteines[16,31], as well as heterobifunctional NHS-diazirine cross-linkers[32]. The MS-labile C-S bonds adjacent to the sulfoxide in these linkers have proven robust and reliable in yielding characteristic fragmentation upon collision-induced dissociation (CID), permitting unambiguous identification of cross-linked peptides[3].

Here, we employ sulfoxide-based cleavability to establish a cysteine-enabled cleavable XL-MS platform, permitting proteome-wide PPI analysis using non-cleavable heterobifunctional K-C linkers. The effectiveness of the established platform is demonstrated using three commercially available and non-cleavable NHS ester-haloacetamide heterobifunctional cross-linkers (Fig. 1a) to map in vivo PPIs from HEK 293 cells. The oxidation-induced cleavability of cross-linked cysteines allows accurate identification of K-C cross-linked peptides using the same LC MS[n] analysis workflow that has been previously established for sulfoxide-containing MS-cleavable cross-linkers[3,4]. To complement MS[n]-based data acquisition, MS[2] analysis using stepped-HCD is also performed[33]. Together, these analyses result in a comprehensive XL-PPI dataset containing 25,401 unique K-C linkages from 2007 proteins, significantly expanding current XL-proteomes. In addition, we perform AlphaFold prediction of intra- and inter-protein interactions to facilitate the interpretation of XL-MS data as well as structural analysis of the proteome. This work represents an XL-MS analysis of the human interactome using heterobifunctional cross-linkers. The platform established here has augmented our capability to map endogenous PPIs. Importantly, the cysteine-based cleavability presents an avenue for developing chemical reagents to further advance XL-MS technologies and benefit chemical proteomics for understanding cellular functions.

## Results

### Developing a cysteine-based MS-cleavable XL-MS technology

In order to expand PPI coverage and improve structural analysis, we aimed to develop an MS-cleavable XL-MS platform using heterobifunctional K-C cross-linkers to complement existing homobifunctional reagents. In previously developed sulfoxide-containing MS-cleavable cross-linkers (e.g. DSSO)[3,27,32,34], the sulfoxide group is introduced within the linker region, yielding MS-cleavable C-S bonds. Here, instead of integrating a sulfoxide into the linker, we attempted to make cysteine cleavable by leveraging its unique chemical properties. This is based on the fact that cysteines can be carbamidomethylated by haloacetamides and subsequently oxidized to introduce a sulfoxide group with an MS-cleavable site[35,36]. Thus, we aimed to develop a cysteine-based cleavable XL-MS platform to facilitate cross-link identification using commercially available, non-cleavable K-C cross-linkers (Fig. 1). To confirm that oxidized carbamidomethylated cysteines yield

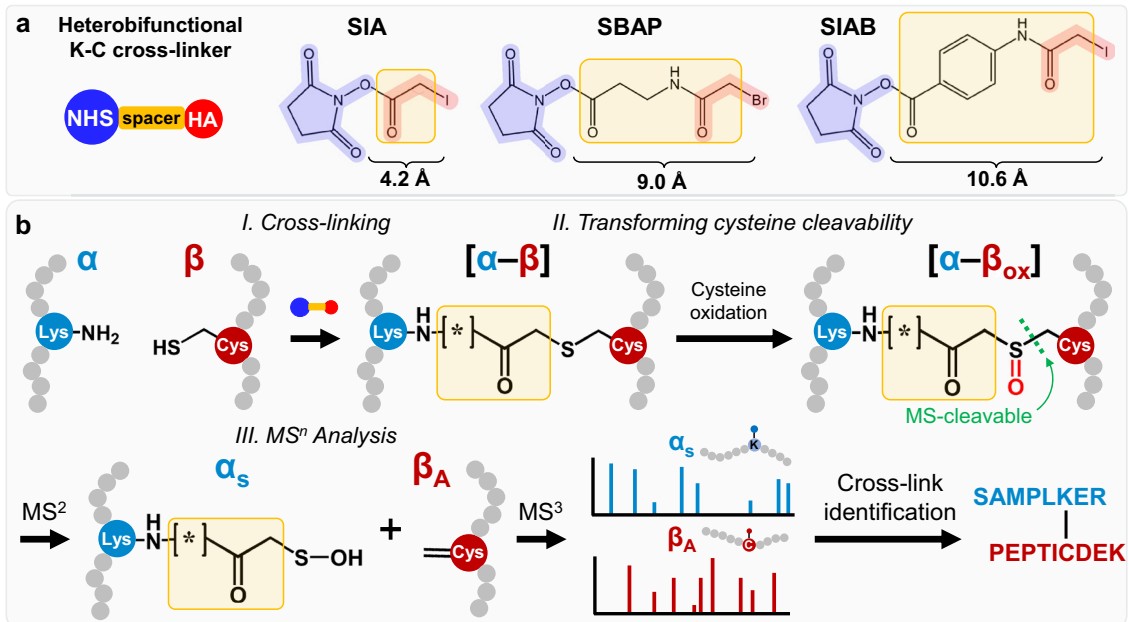

**Fig. 1 | The cysteine-based cleavable XL-MS Technology. a** Structures of the three selected non-cleavable NHS ester-haloacetamide heterobifunctional cross-linkers: SIA, SIAB, and SBAP. Note: NHS: NHS ester; HA: haloacetamide. **b** Three main steps involved in the analysis of K-C cross-linked peptides by MS[n]. After cross-linking (step 1), the resulting K-C cross-linked peptide (α-β) undergoes selective oxidation to introduce a MS-cleavable C-S bond at the cross-linked cysteine, transforming a non-cleavable K-C cross-linked peptide (α-β) to MS-cleavable (Step 2). The sulfoxide-based MS-cleavability enables the generation of characteristic fragments (αS/βA) during collision induced dissociation for subsequent MS[n] analysis (step 3). Note: αS: modified with sulfenic acid (S); βA: modified with alkene (A).

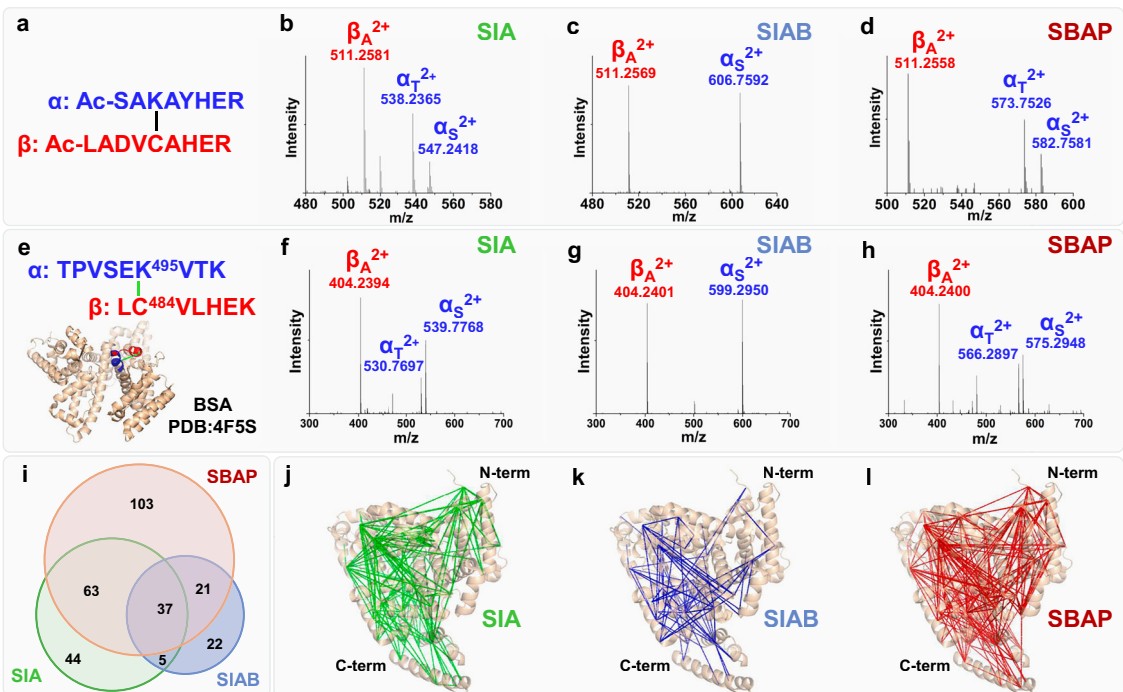

**Fig. 2 | Fragmentation Characteristics of K-C Cross-linked Peptides and XL-MS Analysis of BSA. a** The cross-linked peptide signifying a K-C linkage between synthetic peptides (SR8($\alpha$)-LR9 ($\beta$)) identified by all 3 cross-linkers. **b** MS² analysis of the SIA cross-linked peptide (m/z 529.2494⁴⁺) generated a peptide fragment pair $\alpha_S/\beta_A$ (m/z 547.2418²⁺/511.2581²⁺). **c** MS² analysis of the SIAB cross-linked peptide (m/z 559.0103⁴⁺) produced a peptide fragment pair $\alpha_S/\beta_A$ (m/z 606.7592²⁺/511.2569²⁺). **d** MS² analysis of the SBAP cross-linked peptide (m/z 547.0079⁴⁺) yielded a peptide fragment pair $\alpha_S/\beta_A$ (m/z 582.7581²⁺/511.2558²⁺). $\alpha_T$ (modified with unsaturated thiol moiety (T)) was also detected in (**b**, **d**) due to dehydration of its corresponding $\alpha_S$. **e** The cross-linked peptide signifying a linkage between K495 and C484 of BSA was identified by the three linkers, which were mapped to the BSA structure (PDB: 4F5S). MS² spectra of the representative cross-linked peptide of BSA by (**f**) SIA, (**g**) SIAB and (**h**) SBAP, displaying expected fragmentation patterns. **i** Overlap of K-C linkages identified using the three linkers. 3-D XL-maps of BSA revealed by (**j**) SIA, (**k**) SIAB, and (**l**) SBAP cross-linking.

an MS-labile bond, a cysteine-containing synthetic peptide Ac-LR9 was alkylated by iodoacetamide followed by chloramine-T-induced oxidation[36,37] (Supplementary Fig. 1). As shown, 1-min oxidation was near completion, and this condition has been employed in this work. Collision-induced dissociation (CID) analysis of oxidized and alkylated Ac-LR9 triggered a neutral loss of 107 Da (Supplementary Fig. 1)—corresponding to the loss of RSOH (R = CH₂CONH₂, carbamidomethyl)—and converting the oxidized carbamidomethylated cysteine into hydroalanine, as previously reported[35] (Supplementary Fig. 1). Based on the result, we have designed a general XL-MS workflow in which non-cleavable K-C cross-linked peptides are converted to MS-cleavable ones through oxidation, yielding a structure mimicking our previously developed sulfoxide-containing MS-cleavable cross-linkers[3,27,32]. Cleavage of the single MS-cleavable C-S bond produces two characteristic peptide fragments (i.e. $\alpha_S/\beta_A$) that are subjected to subsequent MS³ analysis (Fig. 1b). The lysine-containing $\alpha$ peptide fragment is modified with a sulfenic acid (S) moiety, while the cysteine-containing $\beta$ fragment is modified with an alkene (A) moiety. Integration of MS¹, MS², and MS³ allows accurate identification of K-C cross-linked peptides using our previously established MSⁿ workflow[3,16,27,32,34].

To demonstrate the effectiveness of the cysteine-based cleavable XL-MS technology, we have chosen three commercially available and non-cleavable K-C cross-linkers that carry different lengths, spacer arms, and haloacetamides: SIA (succinimidyl iodoacetamide, 4.2 Å), SIAB (succinimidyl (4-iodoacetyl)aminobenzoate, 10.6 Å) and SBAP (succinimidyl 3-(bromoacetamido)propionate, 9.0 Å) (Fig. 1a). To evaluate the workflow, we first cross-linked two synthetic peptides: a lysine-containing peptide Ac-SR8 and a cysteine-containing peptide Ac-LR9. Prior to MS analysis, non-cleavable SIA, SIAB, and SBAP cross-linked peptides (SR8-LR9) were oxidized to become MS-cleavable cross-linked peptides, which all displayed expected MS² fragmentation

(Fig. 2a–d). Notably, the sulfenic acid moiety in SIA and SBAP cross-link fragments underwent dehydration to become more stable thiol fragments—similar to sulfoxide-containing MS-cleavable cross-linked peptides[27] (Fig. 2b, d). In contrast, the sulfenic acid-modified fragment ($\alpha_S$) from the SIAB cross-linked peptide was less prone to dehydration, most likely due to stabilization by the benzene group (Fig. 2c). MS³ analyses of the $\alpha_{S/T}$ and $\beta_A$ fragments verified a K-C linkage between SR8 and LR9 peptides by the three selected linkers, respectively (Supplementary Fig. 2a–c). These results demonstrate the feasibility of introducing an MS-labile bond to previously non-cleavable K-C cross-links, thus enabling accurate identification of K-C cross-linked peptides by MSⁿ analysis.

## XL-MS analysis of standard proteins

To examine the K-C XL-MS workflow for protein analysis, BSA was cross-linked, digested, oxidized, and analyzed. Cross-link identification is exemplified by the MSⁿ analyses of representative SIA, SIAB, and SBAP cross-linked peptides signifying a cross-link between K495 and C484 of BSA (Fig. 2e). As illustrated, these cross-links have yielded characteristic fragmentation to allow their MSⁿ-based identification (Fig. 2f–h, Supplementary Fig. 3). From three biological replicates, LC-MSⁿ analyses resulted in a total of 149, 85, and 224 unique K-C linkages identified for SIA, SIAB and SBAP respectively, and no decoy hits were found (Fig. 2i, Supplementary Data 1a–c). The XL-data reproducibility across three biological replicates for each cross-linker ranged from 27.1% to 41.9% (Supplementary Fig. 4). The limited overlap (12.5%) of the K-C linkages among the three linkers (Fig. 2i) suggests their ability to capture complementary PPIs. In comparison to our published XL-MS data of BSA using other residue-targeting chemistries[16,30,31], K-C cross-linking captured more unique intramolecular interactions in the N- and C-terminal regions, further expanding the XL-PPI map of BSA

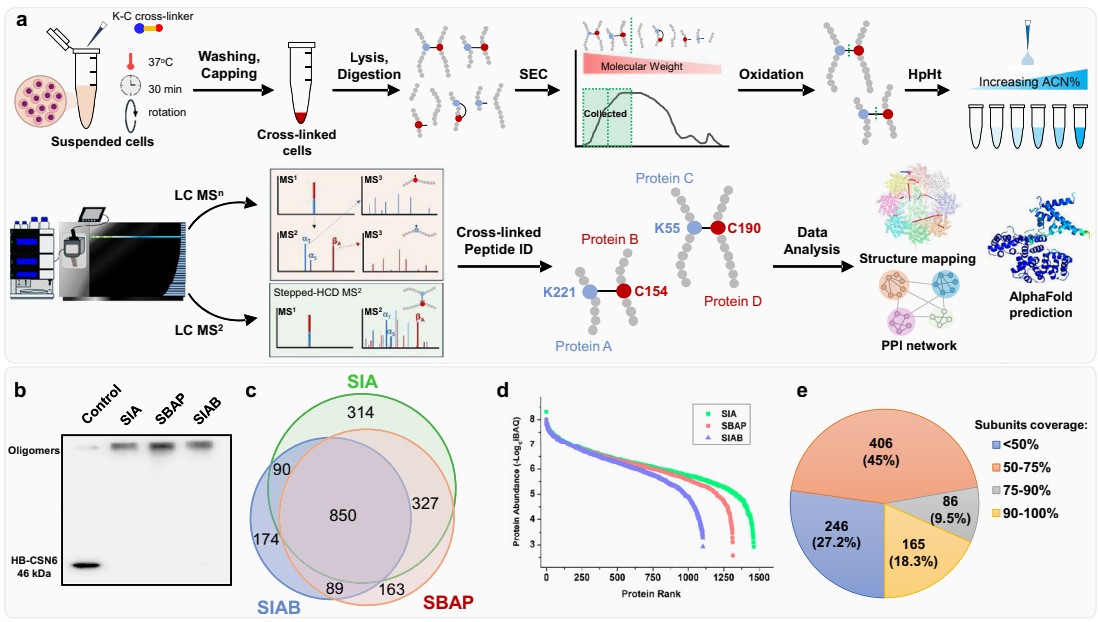

**Fig. 3 | XL-MS analysis of HEK293 cells. a** The general cysteine-based cleavable XL-MS workflow for intact cells. **b** Immunoblotting analysis of cross-linked HEK293 cells by probing HB-tagged CSN6. The experiment was independently repeated two times with similar results. **c** Overlaps of K-C cross-linked proteins identified from SIA, SIAB and SBAP cross-linking. **d** Abundance distribution of the 1856 proteins found in the K-C XL-proteome that have iBAQ values in ProteomicsDB. **e** Distribution of protein composition coverage of CORUM protein complexes identified by K-C XL-MS analyses.

(Fig. 2j–l). When the identified K-C cross-links were mapped to the structure of BSA (PDB: 4F5S), their median Cα-Cα distances were found to be 25.3 Å (SIA), 26.5 Å (SBAP), 26.0 Å (SIAB) respectively (Supplementary Data 1 and Supplementary Fig. 4d). Based on their spacer arm lengths, while SIAB (10.6 Å) and SBAP (9.0 Å) were anticipated to capture cross-links of longer, yet comparable distances, SIA (4.2 Å) was expected to capture cross-links about 5 Å shorter. However, XL-MS data of BSA indicated that SIA was able to capture cross-links with distances similar to those obtained using SIAB and SBAP. The detection of the unexpected cross-links may be associated with the extensive disulfide bonds in BSA and their dynamic rearrangement. To eliminate the interference from disulfide bonds, we performed XL-MS on another standard protein, rabbit aldolase, which contains 8 cysteines but no disulfide bonds. This resulted in a total of 29 SIA, 37 SIAB, 28 SBAP K-C cross-links (Supplementary Data 2a–c, Supplementary Fig. 5a–c). When these cross-links were mapped to a high-resolution structure of aldolase (PDB:8EW2), their median Cα-Cα distances were determined as 23.4 Å (SIA), 25.2 Å (SBAP) and 26.8 Å (SIAB), comparable to those of BSA cross-links. In addition, the overall cross-link distance distributions for both BSA and aldolase were not significantly impacted by these linkers' lengths (Supplementary Figs. 4d, 5d). While SBAP and SIAB behaved as expected, SIA displayed the capability to capture longer cross-links than expected, independent of its linker length. This observation is most likely attributed to the unique chemical properties and high reactivity of SIA. Regardless, these results have demonstrated that our cysteine-based cleavable XL-MS workflow is robust, versatile, and applicable for non-cleavable K-C cross-linkers with different functionalities.

**In vivo XL-MS analysis of HEK 293 cells**
To determine whether the selected K-C cross-linkers can be employed to expand the coverage of cellular networks, we have established a general in vivo XL-MS workflow (Fig. 3a). First, membrane permeability and in-cell cross-linking efficiency of the three selected K-C linkers were confirmed by immunoblot analysis (Fig. 3b). Next, proteins were extracted from cross-linked cells and digested, and cross-linked peptides were enriched by peptide SEC followed by cysteine oxidation and

high pH reverse phase tip (HpHt) fractionation prior to LC MSⁿ analysis[15]. The MSⁿ-based analyses generated a total of 66,989 K-C cross-linked peptides from 1612 proteins in HEK 293 cells, with average FDRs of 0.07%, 0.1%, and 0.03% at the CSM (cross-link spectrum match) level for SIA, SIAB, and SBAP, respectively, across biological replicates (Supplementary Fig. 6a, d, Supplementary Data 3). In addition, we performed MS²-based analysis for 2 SIA replicates, 3 SIAB replicates and 3 SBAP replicates (Supplementary Methods) to increase the depth of XL-MS data, identifying 57,059 K-C cross-linked peptides from 1649 proteins with 1% FDR at the CSM level using Protein Prospector (Supplementary Methods and Supplementary Data 4). The integration of LC-MSⁿ and -MS² datasets has revealed a combined XL-proteome composed of 2007 proteins with 25,401 unique K-C linkages (Fig. 3c, Supplementary Data 5).

To understand the depth of the XL-proteome determined by the three selected K-C linkers, the abundances of the 1856 XL-proteins with corresponding intensity-based absolute quantification (iBAQ) values in ProteomicsDB database were examined. The results indicated that the K-C XL-proteome encompasses proteins with abundances spanning over 7 orders of magnitude (Fig. 3d), comparable to previously published XL-data[17]. Protein complex analysis identified 903 CORUM protein complexes, 72.8% of which had over 50% subunit recovery (Fig. 3e, Supplementary Data 6a). In addition, 5904 K-C linkages were mapped across 573 protein complexes that have high resolution structures in the PDB database (Supplementary Data 6b). Gene Ontology (GO) analyses discovered that extensive cellular compartments, biological processes, and molecular functions were covered by the K-C linkers (Supplementary Fig. 7), illustrating their effectiveness for in-cell cross-linking. Due to differences in both cross-linking chemistry and linker design—including reactive groups, target residue specificity and linker arm lengths—847 cross-linked proteins were uniquely identified in this study (Supplementary Fig. 6b) compared to our previously reported in vivo K-K XL-proteome revealed by DSBSO cross-linking (14 Å)[17]. Unlike DSBSO, which targets lysine-lysine pairs, the three K−C linkers used here have shorter spacer arms and different reactivities, contributing to this increased proteome coverage. When protein interactions are captured using K−K cross-linkers,

identification of all interacting proteins relies solely on cross-linked lysine-containing peptides. In contrast, K−C cross-linkers enable the identification of interaction partners through cross-linked lysine-containing peptides, cysteine-containing peptides, or a combination of both. Interestingly, the majority (1734 out of 2007) of the K-C XL-proteins were identified with both cross-linked lysine-containing and cysteine-containing peptides, however, 68 and 205 proteins were found exclusively with cross-linked cysteines and lysines, respectively (Supplementary Fig. 6c). For those only identified with cross-linked lysines, 82 have not been reported in our previous in vivo K-K XL-data[17]. To understand why these proteins are unique to K-C linkers, we examined two selected interactions in more details. For example, the interaction between EIF4E and EIF4G1 was identified through the cross-link of EIF4E:K192 to EIF4G1:C662, where EIF4E was detected with only cross-linked lysine-containing peptides. This cross-link was reproducibly identified by all three K-C linkers, supporting its validity. Notably, although eight lysines are distributed within residues 600–720 of EIF4G1, none were found to be cross-linked to EIF4E in previous XL-MS studies using lysine-reactive homobifunctional linkers[13,15,17,18,38]. Analysis of the existing (PDB: 5T46) and AlphaFold3 predicted structures of the human EIF4E-EIF4G complex revealed that six of these lysines in EIF4G1 are located 30 - 40Å from K192 of EIF4E, while the remaining two lysines are much farther away (85.1 Å and 86.6 Å) (Supplementary Fig. 8a, b). Interestingly, this EIF4E-EIF4G1 interaction was captured exclusively using K-C cross-linking, made possible by the proximity of EIF4G1:C662 to EIF4E:K192 (10.3 Å).

Another example involves the interaction between S100A10 and ANX2 through the identification of S100A10:K47–ANXA2:C9 cross-link. Based on the AlphaFold3 predicted structure of the S100A10-ANXA2 complex, both K10 (11.4 Å) and K28 (21.2 Å) of ANXA2 are close to K47 of S100A10 and thus their cross-links should be observed using existing lysine cross-linkers (Supplementary Fig. 8c). Despite their proximity, ANXA2:K10 and ANXA2:K28 have not been found cross-linked to S100A10:K47 in previous studies[13,15,17,18,38]. This is most likely due to the detectability of the resulting tryptic peptides. In this work, cross-linking of ANXA2:C9 to S100A10:K47 involved the identification of a 10-residue long tryptic peptide (MSTVHEIL**C**K) of ANXA2. Formation of an ANXA2:K10 or ANXA2:K28 cross-link to S100A10:K47 using lysine-specific cross-linkers would both result in much longer (28-residue) peptides and potentially impeding cross-link detectability. These examples suggest that the exclusive capture of these proteins' interactions by K−C linkers is most likely attributed to the availability and accessibility of lysine-cysteine pairs at interaction interfaces, as well as the detectability of resulting cross-linked peptides. Collectively, these results indicate that heterobifunctional K-C cross-linkers are capable of capturing diverse PPIs in cells to complement existing homobifunctional cross-linking reagents and expand the coverage of the XL-proteome.

## Distance mapping of intra-protein K-C cross-links onto existing structures

To better assess the XL-data, we have performed structural analysis of intra- and inter-protein cross-links separately, using only reproducible cross-links that were identified from at least two biological replicates with the same cross-linker. Based on the 5727 unique intra-protein K-C cross-links identified here, we found an average of 4.5 cross-links per protein with a median of 2.0 (Supplementary Fig. 9a), including significant overlap between the three K-C cross-linkers (Supplementary Fig. 9b). Among the 5727 intra-protein cross-links, 3428 were mapped to PDB structures of 701 proteins, monomers and homo-oligomers (Fig. 4a). We found that 76% of the mapped cross-links were satisfactory in monomeric structures (Cα-Cα distance <30 Å), supporting their validity (Fig. 4b, Supplementary Data 5). To examine whether any of the violating intra-protein cross-links could represent inter-subunit cross-links from homomeric oligomers, we computed the Cα-Cα

distances between cross-linked residues from adjacent proteins within available homo-oligomeric structures. As a result, an additional 92 violating intra-protein cross-links were satisfied when mapped to homomeric interactions, increasing the overall satisfaction rate to 79% (Supplementary Data 5). For example, the monomer of the aldolase satisfied 35 out of 42 cross-links, while 3 additional cross-links were satisfied when the tetramer was considered (Fig. 4e). To further understand the occurrence of these cross-links, we examined whether cross-link distance satisfaction correlated with their identification frequency across different linkers. However, no significant correlation was observed (Supplementary Fig. 9d), suggesting that additional conformational states have yet to be resolved. Since rabbit aldolase shares high sequence identity with the human protein (~95%), we were able to directly compare the in-cell data with the in vitro analysis of the standard protein described earlier. In comparison, a 58.5% overlap was observed among the K−C cross-links identified by all linkers across both conditions (Supplementary Fig. 10a), demonstrating the consistency of our XL-MS workflow. Notably, the Cα-Cα median distances of in-cell cross-links were smaller than in vitro cross-links (Supplementary Fig. 5d, Supplementary Fig. 10b), implying that the human aldolase cross-links mapped better to its high-resolution structure than those from rabbit aldolase in solution.

## Distance mapping of intra-protein cross-links onto AlphaFold2 structures

To examine additional possible conformations, we have applied AlphaFold2 (AF2)[39] to proteins with PDB structures but unsatisfied cross-links. Since the stoichiometry of homomeric structures is unknown, we used AlphaFold-Multimer v2.3 (AFM)[40] to predict dimers. This led to the satisfaction of an additional 68 cross-links in 50 proteins where AF2 produced a conformation different from the PDB structure (Supplementary Data 5). For example, the HSP70 conformation produced by AFM slightly moves one of the domains, resulting in the satisfaction of 3 additional cross-links (Fig. 4f). In total, 81% of cross-links mapping to PDB structures were satisfied if we consider AFM conformations (Fig. 4a). Next, we applied AFM to predict structures for proteins without structural coverage in the PDB. For monomeric AFM structures, the fraction of satisfied cross-links (Cα-Cα distance <30 Å) was 63% (1276/2017 cross-links). When considering the homodimers predicted by AFM, an additional 61 cross-links derived from 46 proteins were satisfied, resulting in a satisfaction rate of 66% (Supplementary Data 5). However, the satisfaction rate remains significantly lower than the 81% satisfaction rate observed with PDB structures (Fig. 4a, Supplementary Data 5). One such example is the GDI1 protein with 6 out of 10 satisfied cross-links (Fig. 4g). We attribute this discrepancy primarily to the lower accuracy of the AFM structures. To test this hypothesis, we analyzed the relationship between AFM confidence metrics and cross-link satisfaction. We compared the Predicted Aligned Error (PAE) metric of cross-linked residues to their corresponding Cα-Cα distances and found that cross-links were satisfied in most low-scoring PAE structures, suggesting a correlation between low PAE scores and increased cross-link satisfaction (Fig. 4d). By setting a confidence threshold of PAE < 20 for an AFM model to be considered, the satisfaction rate increased to 79% (1189/1506) (Fig. 4a, c). We obtained similar results for the pLDDT confidence measure (Supplementary Fig. 9c). These results suggest that cross-links can be used alongside AFM confidence metrics to provide orthogonal information on model accuracy.

To account for the influence of protein dynamics on the satisfaction rate, we examined whether cross-link distance satisfaction correlates with identification frequency across different linkers and biological replicates. We assume that cross-links observed in multiple replicates and linkers are more likely to originate from structurally stable regions. Indeed, cross-links identified by more than one linker also showed shorter median distances (Supplementary Fig. 9d).

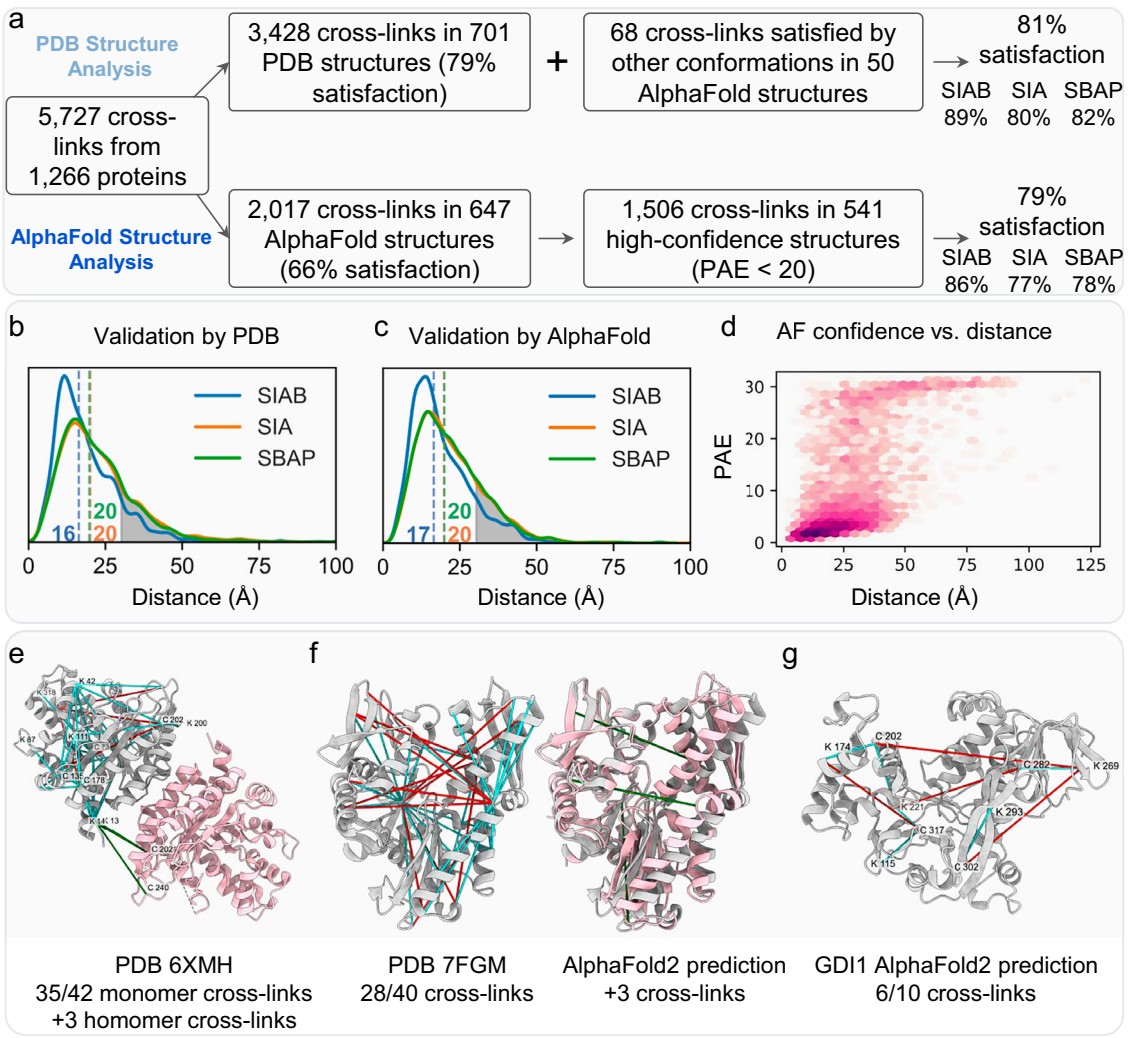

**Fig. 4 | Structural analysis of intra-protein cross-links. a** Mapping cross-links to PDB and/or AlphaFold structures. In the first track, cross-links were first mapped to available PDB structures. When a cross-link was not satisfied, AlphaFold was used to search for additional conformations that may satisfy it. In the second track, cross-links in residues not present in any PDB structure were mapped to AlphaFold2 predicted structures and only cross-links between residues with high confidence were considered. **b, c** Distribution of distances between cross-linked amino acids in the PDB structures (**b**) or in high-confidence AlphaFold2 structures (**c**). **d** Distance between cross-linked amino acids vs. Predicted Aligned Error (PAE). **e** K-C cross-links mapped to aldolase tetramer (PDB 6XMH, only dimer is shown for clarity).

Cross-links were depicted as lines - cyan for cross-links satisfied in the same chain, green for cross-links satisfied by the homomeric interaction, and red for violated cross-links. **f** K-C crosslinks mapped to the HSP70 - an experimental structure (left, silver, PDB 7FGM), and an AlphaFold2 prediction (pink) superimposed on the experimental structure (right). The experimental structure satisfied 28 out of the 40 cross-links (cyan) and the slightly different conformation predicted by AlphaFold2 satisfied three additional cross-links (green). **g** AlphaFold2 structure for GDP dissociation inhibitor alpha, which does not have any experimental structures with cross-links mapped to it. Six cross-links were satisfied (cyan) and four were violated (red).

---

Likewise, cross-links detected more frequently across biological replicates exhibited lower median distances (Supplementary Fig. 9e–g) and higher satisfaction rates (Supplementary Fig. 9h), further supporting the notion that reproducibility is associated with structural stability.

**Structural analysis of inter-protein K-C cross-links**

In this work, we obtained 8690 unique inter-protein K-C cross-links that were reproducibly identified from all three cross-linkers, yielding 3031 PPIs with an average of 2.9 cross-links per PPI and a median of 2.0 (Supplementary Fig. 9i). However, structural coverage was only available for 185 cross-links and 82 PPIs (Supplementary Fig. 9j). In contrast to intra-protein cross-links, the majority of inter-protein cross-links were non-satisfactory (47% <30 Å, Fig. 5a, Supplementary Data 5). 38% of the cross-links (71 out of 185) with structural coverage were between ribosomal proteins. Of these, only 11 were satisfied (15%), suggesting that many non-satisfactory cross-links most likely correspond to

interactions between neighboring ribosomes located on the same mRNA molecule. For non-ribosomal cross-links, the satisfaction rate was 67%. Notably, unlike the intra-protein cross-links, the satisfaction rate of inter-protein cross-links appeared to correlate with their identification frequency by different cross-linkers. For cross-links that were only observed by all three cross-linkers, the satisfaction rate increased to 75% (15/20, Fig. 5b). Additionally, we applied AFM to predict alternative conformations for several protein interactions, resulting in the satisfaction of 3 additional cross-links across 2 PPIs. These results suggest that existing structures represent the most prominent and stable interactions within a protein complex and that inter-protein interactions are much more dynamic than intra-protein interactions.

Next, we applied AFM to predict structures for PPIs without structural coverage and computed $C\alpha$–$C\alpha$ distances between cross-linked residues only from high-confidence models (cross-linked residues with PAE < 20) (Fig. 5c, Supplementary Fig. 9k). There were 1303 cross-links with PAE < 20, covering 343 PPIs with a satisfaction ratio of

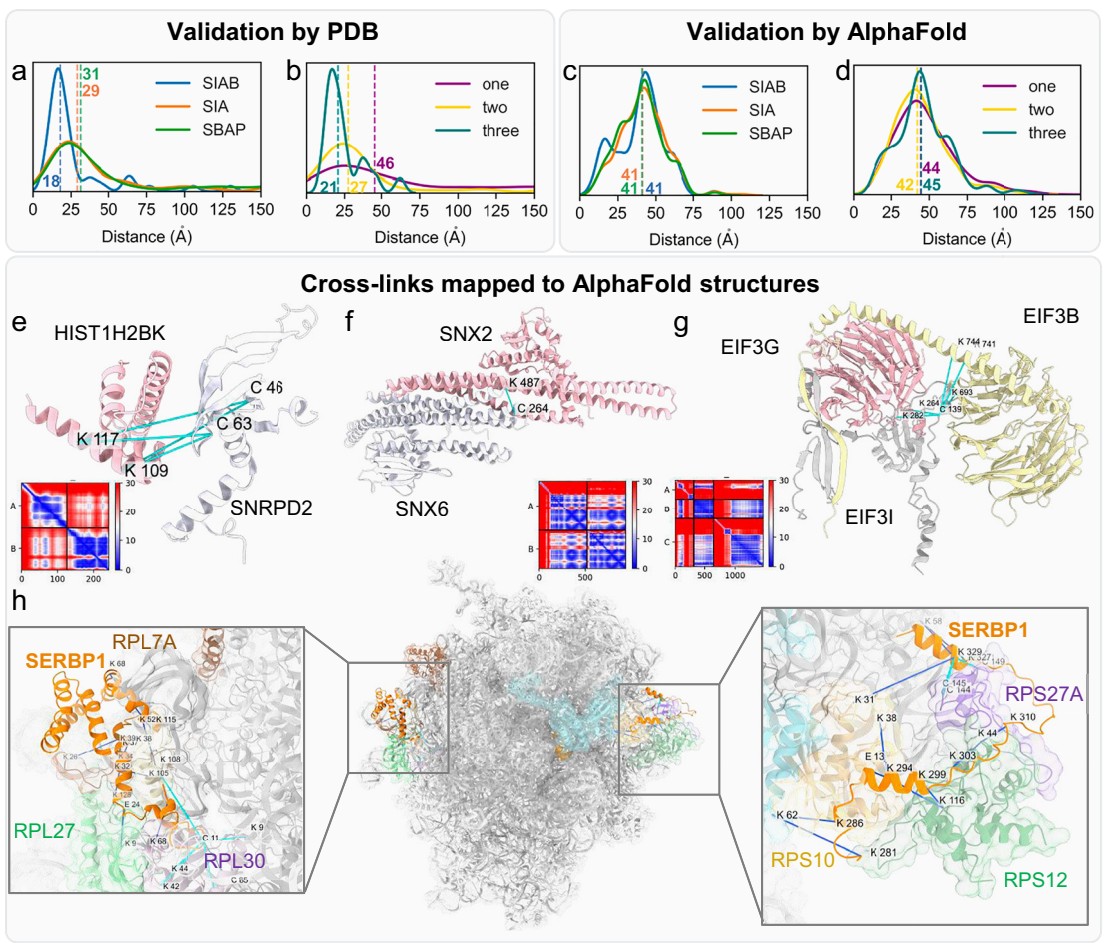

**Fig. 5 | Structural analysis of inter-protein cross-links. a, b** Distribution of distances between cross-linked amino acids mapped to PDB structures (**a**) for each of the three cross-linkers and (**b**) according to the number of the cross-linker types. **c,d** Distribution of distances between cross-linked amino acids mapped to Alpha-Fold structures (**c**) for each of the three cross-linkers and (**d**) according to the number of the cross-linker types. **e–g** Complexes without experimental structures, in which the AFM prediction confidence and the K-C cross-links mapped to the structure agree with each other. Next to each structure, there is the AFM Predicted Aligned Error (PAE) between pairs of residues which deemonstrates the confidence of AFM in the structure. **e** HIST1H2BK (pink) and SNRPD2 (silver) with four cross-links (the 5th cross-link maps to a disordered fragment and not shown here). **f** SNX2 (pink) and SNX6 (silver) with one cross-link. **g** EIF3G (pink), EIF3I (silver), and EIF3B (gold) with five cross-links. **h** SERBP1 N-terminal fragment (residues 1–140, left) and C-terminal fragment (residues 280–339, right) wrap the ribosome on the two sides consistent with K-C (cyan) and K-K (blue) cross-links.

27% (Cα-Cα distance <30 Å). Cross-linked residues identified by two or three K-C linker types are more likely to have a PAE < 20 (Supplementary Fig. 9l), but the distance distribution was not significantly impacted (Fig. 5d). Several factors may contribute to this low satisfaction rate. First, the success rate of AFM on benchmarks for estimating PPI prediction accuracy is 60% while AlphaFold2 reaches 90% success rate for monomeric proteins[41]. Second, the assumption of 1:1 stoichiometry during AFM analysis might be incorrect, potentially impacting prediction accuracy. Third, AFM's success rate is lower when predicting full-length proteins compared to protein fragments, such as interacting domains[42].

### Examination of the K-C XL-PPIs
To further illustrate the human interactome, we constructed a K-C XL-PPI network containing 2007 proteins and 6589 interactions (Supplementary Fig. 11a), and compared it to previous in vivo and in vitro XL-MS data of HEK293 cells using lysine and cysteine homobifunctional cross-linkers[15–17] and the selected PPI databases (BioPlex and BioGRID). As a result, 13.8% (670) of the K-C XL-PPIs were found in published XL-MS studies, 11% (536) and 5.8% (283) were reported by the BioGrid and BioPlex databases, respectively, leaving 3291 PPIs that have not been previously reported (Supplementary Fig. 11b), increasing the overall

interaction coverage of cellular networks. 2446 out of 4872 PPIs were identified by multiple K-C linkers, suggesting the reliability of their identification. Of the 4872 XL-PPIs identified in this study, 559 high-confidence interactions—derived exclusively from proteins identified with unique peptides—could be mapped to the STRING database, with 70.7% of these exhibiting STRING scores greater than 0.9 (Supplementary Fig. 11c), consistent with previous XL-MS studies and underscoring the robustness of our dataset[15–17]. Moreover, we coupled cross-link satisfaction with AFM predictions to examine PPIs that do not have STRING scores. As a result, 166 additional PPIs were validated by interprotein cross-links consistent with high-confidence AFM models, and 157 of them involved interactions with ribosome chains or histones. This is exemplified by the interaction between HIST1H2BK and SNRPD2, which was supported by an AFM model and 80% (4/5) cross-link satisfaction (Fig. 5e). Furthermore, we analyzed known PPIs without experimentally solved structures and found 137 interactions with high-confidence AFM models and satisfied cross-links. For instance, it is known that heterodimers can be formed between sorting nexin-1/2 (SNX1/SNX2) and sorting nexin-5/6 (SNX5/SNX6)[43] proteins, however their interactions have not been revealed by previous K-K XL-MS studies. Here, our K-C XL analysis revealed the interaction between SNX2 and SNX6 with a cross-link satisfied in the AFM model (Cα-Cα distance

of 8Å) and an interface PAE of 4 (Fig. 5f). Finally, we examined the interactions within the eIF3G-eIF3I-eIF3B trimer, which is part of a larger eIF3 complex, with only partial structure available for S. cerevisiae (PDB 4U1E)[44]. The AFM model is validated by five K-C cross-links with Cα-Cα distance <30Å and an average interface PAE of 6.2 (Fig. 5g). These results indicate that the integration of XL-MS data and AFM analysis permits validation of both novel and known PPIs with structural features and higher confidence. This aligns with previous studies highlighting the benefits of AlphaFold for XL-MS-based structural modeling.

### Complementarity of cross-linking chemistry

Based on the K-C XL-data obtained here, it is evident that cross-linkers with different functionalities can yield unique PPIs, producing supporting and/or complementary information to known interactions. For example, while intra-protein interactions within SFPQ and NONO domain-containing octamer-binding protein were determined by both the K-K (DSBSO) and K-C linkers (SIA, SIAB, SBAP), their inter-protein interaction was only uncovered by K-C cross-links, revealing an unknown PPI (Supplementary Fig. 12a). Interestingly, the interaction between Rpt1 and Rpt2 was determined by both DSBSO (Rpt1:K46-Rpt2:K61) and K-C cross-linkers (Rpt1:K34-Rpt2:C58; Rpt1:K58-Rpt2:C58), in which the residues cross-linked by the two different reagents were located in close proximity, reaffirming the identified interaction contacts (Supplementary Fig. 12b). Notably, some XL-PPIs were preferably identified by specific cross-linking chemistries. For instance, the interaction between CCT2 and CCT5 that was identified by 15 K-C linkages here was only determined by a single K-K cross-link previously[17]. Interestingly, we found two cross-links that were conformation-specific, one satisfied only in the closed conformation and the other in the open (Supplementary Fig. 12c). This suggests that K-C cross-linkers are better suited for dissecting the CCT2-CCT5 interaction. Other examples include the interaction between CDC123 and EIF2S3, which was only identified by K-C (Supplementary Fig. 12d) and C-C cross-linkers[16] but not K-K cross-linkers[15,17], indicating that cysteine-reactive cross-linkers are preferable for capturing the CDC123-EIF2S3 interaction. While the EIF3G-EIF3I-EIF3B complex was identified with five K-C cross-links (Fig. 5g), only one DSBSO K-K cross-link was found between EIF3I and EIF3B (Supplementary Fig. 13a). In the aforementioned HIST1H2BK-SNRD2 complex, only one K-K cross-link was obtained by DSBSO that was not satisfied in the AFM model (Supplementary Fig. 13b).

To further compare K-K and K-C linkages, we calculated the secondary structure preferences for their satisfied cross-linked positions. We find that lysine residues in the K-C linkages have a preference for helical secondary structure, similar to the K-K linkages that have a preference for helices and loops. In contrast, the cross-linked cysteines are more likely to be part of the beta-sheet (Supplementary Fig. 14a). We also analyzed the distribution of Cα-Cα distances based on secondary structures. We found that when one or both cross-linked residues belonged to a loop region, the distances were larger (Supplementary Fig. 14b). Collectively, our results demonstrate that K-C cross-linker-based XL-MS analyses have yielded additional molecular details to support known interactions and uncover interactions to complement existing K-K linkers, expanding the PPI coverage of cellular networks.

### Cross-link assisted modeling to elucidate the structures of protein complexes

The cross-linking complementarity was further exemplified by the interactions between SERBP1 and the ribosomal complex. Based on 26 K-C cross-links, SERBP1 was found in close contact with 8 ribosomal subunits (RPL30, RPS11, RPS19, RPS27, RPS27A, RPL27, RPL34, RPL36, Supplementary Fig. 15). While most of the SERBP1 interactions have been revealed by other cross-linking chemistries[7,13,15–17], its interactions

with RPS11 and RPS27 were only determined by K-C cross-links. SERBP1, a disordered protein, binds in the mRNA entry channel to prevent mRNA from binding at the A and P sites[45]. Despite its functional significance in gene translation, the structure containing full-length SERBP1 and the ribosome complex has not been resolved. Only a short fragment of SERBP1 (residues 157-188) could be resolved in the mRNA entry channel. Therefore, we have applied AlphaFold3[41] to model two complexes based on our K-C and the available K-K and C-C cross-links[7,13,15–17]: (1) SERBP1 N-terminal fragment (residues 1-140) with RPL27, RPL30, and RPL34 and (2) SERBP1 C-terminal fragment (residues 285-330) with RPS3, RPS27A, RPS12, and RPS10. The resulting predictions were consistent with 40% of inter-protein cross-links and could be superimposed to the full ribosomal structure without steric clashes (Fig. 5h). These models reveal additional SERBP1 ribosomal docking sites that are difficult to identify in cryo-EM analysis due to the transient nature of its interactions. The relatively low satisfaction rate is expected for this dynamic interaction where multiple conformations are likely to explain a larger fraction of the data.

To further illustrate the utility of our XL-MS data, we next performed a structural analysis of the human translation elongation factors, known as the eEF1 family. This protein family includes two variants of eEF1A (eEF1A1 and eEF1A2) and the eEF1B complex consisting of the eEF1B2, eEF1D, and eEF1G subunits (Fig. 6a). The structure and stoichiometry of the eEF1B complex with and without eEF1A are not well characterized. Experimental structures are only available for partial subcomplexes for eEF1G (residues 1-218) with eEF1D (residues 1–30, PDB 5JPO) or eEF1B2 (residues 1-88, PDB 5DQS) and eEF1A · eEF1B2 (yeast, GEF domain, PDB 1IJE). Structure determination of this complex is challenging due to long-disordered regions connecting the globular domains: eEF1B2 (23%), eEF1D (33%), and eEF1G (12%). To facilitate the modeling of the eEF1 complexes, we employed AlphaFold3 followed by model validation using the 64 K-C cross-links obtained here along with 52 K-K and 4 C-C cross-links previously identified[17]. In total, there were 85 intra-protein and 35 inter-protein cross-links.

First, we applied AlphaFold3 to compute the eEF1D trimer model, describing the core component in the complex assembly[46]. In this model, the GEF domains were in close proximity to the helical trimerization core, and this inter-domain interaction was supported by 3 K-C cross-links (Fig. 6b). Second, we applied AlphaFold3 to produce a subcomplex of eEF1B2-eEF1D-eEF1G with 1:1:2 stoichiometry (Fig. 6d). This stoichiometry was assumed based on the available eEF1G dimer structure with the eEF1D N-terminal fragment (PDB 5JPO). This subcomplex revealed an interaction between the GEF domains of eEF1B2 and eEF1D which was validated by 16 cross-links (Fig. 6a). The GEF domain heterodimer was sandwiched between the two domains of the eEF1G homodimer supported by 4 and 5 satisfied cross-links with eEF1B2 and eEF1D, respectively. The model also accurately reproduced available experimental structures (PDBs 5DQS, 5JPO). Third, three copies of the resulting model were superimposed on the eEF1D trimerization core helices (Fig. 6b) to assemble a complex of eEF1B2-eEF1D-eEF1G with 3:3:6 stoichiometry (Fig. 6f).

The eEF1D-eEF1B2-eEF1G-eEF1A1 complex was modeled similarly. We applied AlphaFold3 to produce a subcomplex of eEF1B2-eEF1D-eEF1G-eEF1A1 with 1:1:2:2 stoichiometry (Fig. 6e). The dimeric interfaces of C-terminal eEF1G domains and the GEF domains are dissociating to allow the binding of eEF1A1 to the GEF domains of both eEF1B2 and eEF1D (Fig. 6d). This model is supported by 4 satisfied cross-links with eEF1B2 and 4 with eEF1G. The final model of the whole complex with a stoichiometry of 3:3:6:6 was computed by superposition of its three copies on the eEF1D trimerization core helices (Fig. 6g). Altogether, our models of eEF1D trimer (Fig. 6b), eEF1B2-eEF1D-eEF1G (Fig. 6f), and eEF1B2-eEF1D-eEF1G-eEF1A1 (Fig. 6g) satisfy 85% of the 120 cross-links (Fig. 6c), revealing previously uncharacterized domain-domain interactions and the dynamic nature of the complex assembly.

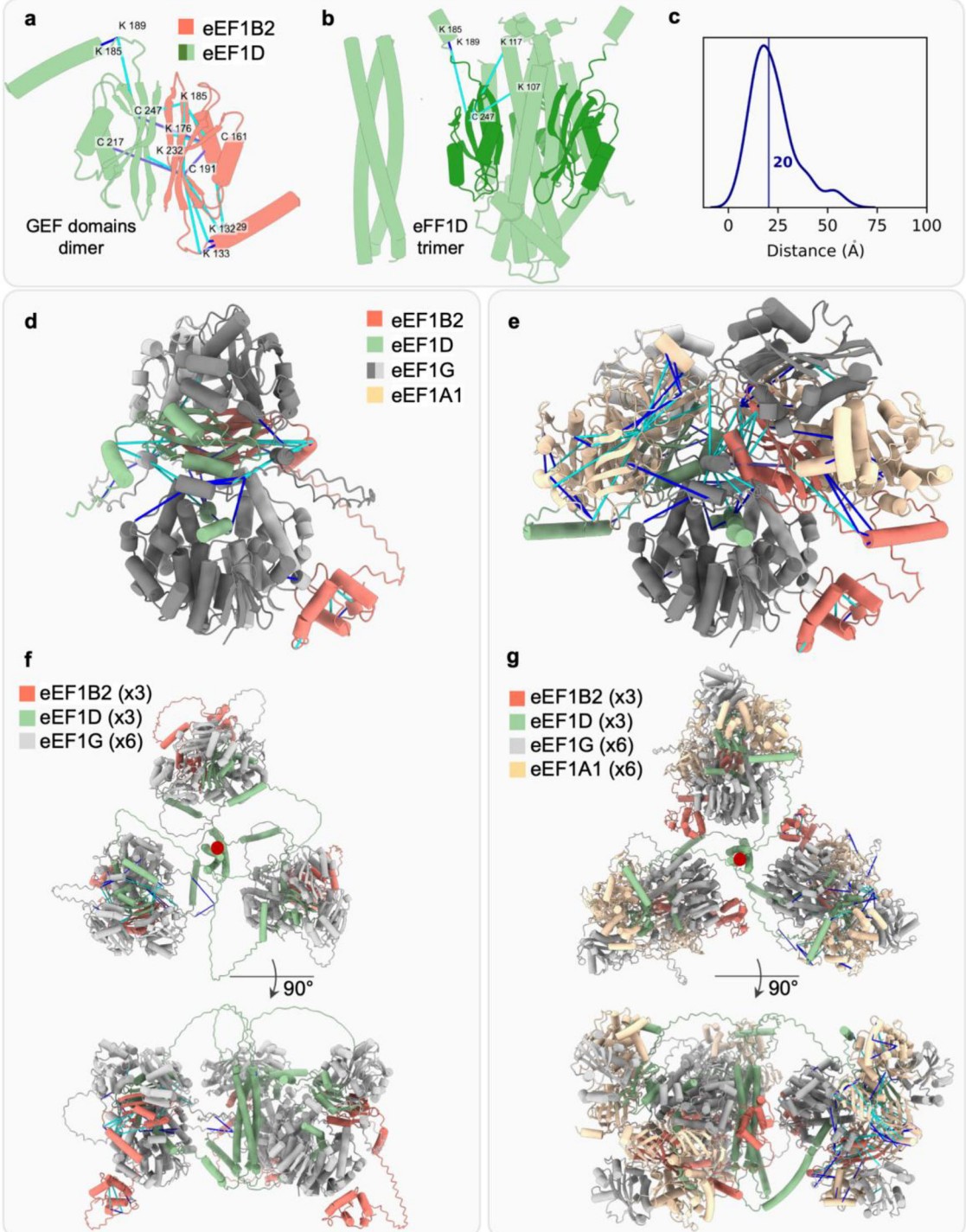

**Fig. 6 | eEF1 complex modeling. a** GEF domains heterodimer: eEF1B2 (coral) and eEF1D (light green) with K-C (cyan), K-K (blue), and C-C (purple) cross-links. **b** Trimerization core of eEF1D (left) and full-length eEF1D trimer (right) with disordered regions not shown for clarity. **c** distances distribution of K-C linkages mapped to eEF1 complex, median distance is indicated by a line. **d, e** eEF1B2-eEF1D-

eEF1G and eEF1B2-eEF1D-eEF1G-eEF1A subcomplexes with supporting cross-links. The trimerization helix and the disordered regions of eEF1D are hidden for clarity. **f, g** Full-length eEF1B2-eEF1D-eEF1G and eEF1B2-eEF1D-eEF1G-eEF1A complexes: top and side views. The eEF1D symmetry axes are indicated in the top view by a red point.

## Discussion

In this work, we have established a cysteine-based MS-cleavable XL-MS analytical platform that enables comprehensive PPI profiling in living cells using commercially available, non-cleavable heterobifunctional K-C cross-linkers. Oxidation-induced cysteine cleavability is robust and allows effective conversion of K-C cross-linked peptides from non-cleavable to cleavable, permitting their unambiguous identification by

$MS^n$ analysis at the systems level. Such transformation has been successfully applied to the three selected K-C cross-linkers (i.e. SIA, SBAP, SIAB) with distinctly different chemical features, demonstrating the robustness of sulfoxide-based MS-cleavability and the general applicability of the developed XL-MS technology. The ability to introduce an MS-cleavable site after cross-linking presents unique opportunities to: (1) take advantage of existing non-cleavable, cysteine-reactive homo-

and hetero-bifunctional cross-linkers; (2) examine the benefits of cross-linker functionality and MS-cleavability; and (3) compare different MS acquisition methods for maximizing cross-link identification in complex samples. In addition, it offers enhanced flexibility for designing cross-linkers, thereby advancing XL-MS studies toward systems structural biology. Notably, because few XL-MS studies have employed asymmetric heterobifunctional MS-cleavable cross-linkers, most existing search algorithms are not yet fully optimized to account for their characteristic fragmentation patterns, such as those produced by SIA, SIAB, and SBAP. In contrast, Protein Prospector enables linker-specific modification settings and has proven effective in identifying K-C cross-links from both MS[2] and MS[3] data. Collectively, this work represents a technological advancement in probing cellular networks and establishes a valuable resource for developing future software tools to improve identification of cross-linked peptides generated by asymmetric heterobifunctional MS-cleavable cross-linkers.

In chemical proteomics, haloacetamides are commonly used for global cysteine reactivity profiling to determine hyper-reactive cysteines[47,48] and studying cysteine oxidation to elucidate cellular redox states[49,50]. Since the oxygen atoms incorporated during cysteine oxidation by chloramine-T are derived from water, an isotope label can be introduced during this process to enable quantitative analysis. Specifically, cysteine-labeled peptides could be oxidized in light ($H_2O^{16}$) and heavy ($H_2O^{18}$) water to generate light and heavy labels, allowing for quantitative comparison of cysteine reactivity or oxidation states without the need for synthesizing isotopically labeled reagents. In addition, due to the low abundance of cysteines, alkyne-tagged haloacteamides are often used to allow click-chemistry-based enrichment by conjugating an affinity tag such as biotin[50,51]. Although such enrichment enhances the detectability of cysteine-containing peptides, the attachment of a large chemical moiety to cysteines after click conjugation could complicate their MS identification. However, by incorporating oxidation-induced cleavability, the covalently attached group can be selectively removed during MS/MS, simplifying peptide analysis and facilitating their accurate identification. Moreover, similar approaches can be applied to accelerate the discovery of haloacetamide-based covalent drugs[52,53]. Therefore, the strategy established here is poised to advance chemical proteomics by improving cysteine identification and characterization, ultimately broadening the impact of this work. Although chloramine-T induced oxidation is highly selective for carbamidomethylated cysteines under our experimental conditions, analysis of BSA XL-data suggested an enhanced oxidation of tryptophan. Including tryptophan oxidation as a variable modification can increase peptide identification; however, its overall impact remains limited given the low abundance of tryptophan residues in the human proteome (~1%). Nonetheless, considering tryptophan oxidation during database searching could be beneficial particularly when analyzing proteins rich in tryptophan.

While the three selected cross-linkers are non-enrichable, we were able to employ a two-dimensional peptide fractionation strategy[15] to improve the detection of K-C cross-linked peptides. This has led to the identification of 25,401 unique linkages from cross-linked HEK293 cells, which allowed us to construct an in vivo interaction map composed of 2007 proteins, complementing the existing XL-proteome. In comparison to BioPlex and BioGRID databases as well as the published XL-data[7,13,15–18], 59.1% of the identified interactions have not been reported before. Furthermore, our results reveal additional physical contacts that confirm known interactions. Due to the variances in their linker regions and/or reactive groups, the three selected linkers appear to have different reactivity, solubility and membrane permeability, resulting in a limited overlap at the residue-to-residue level. Thus, XL-MS analyses employing multiple cross-linkers with the same residue specificity but differing in reactive groups and/or linker structures not only allow cross-validation of PPIs, but also yield extensive interaction contacts to facilitate structural characterization of protein complexes and expand the scope of XL-PPI maps. Altogether, our results demonstrate that K-C heterobifunctional cross-linkers are effective for in-depth XL-MS analysis of cellular networks, complementing existing homobifunctional cross-linking reagents.

While it is generally expected that the spacer arm length of a cross-linker correlates with the distance between cross-linked residues, our data indicate that cross-link satisfaction rates are not strictly determined by this parameter. Although SIA is the shortest linker, its cross-links exhibited Cα−Cα distances comparable to those formed by SBAP and SIAB (Figs. 4, 5). In contrast, SIAB, the longest cross-linker, demonstrated the highest satisfaction rates and lowest median distances for both intra- and inter-protein cross-links in experimental structures and AF models, most likely due to its relatively rigid structure, low reactivity and poor water solubility compared to the other two linkers. Apart from the observed differences in chemical/physical properties and reactivity of these linkers, the dynamic behavior of cysteines may play a significant role in shaping cross-linking outcomes. In comparison to lysine, cysteine's reactivity is highly sensitive to its microenvironment, including local pKa and redox state, and this sensitivity can contribute to longer or unexpected cross-link distances. Based on our in-cell K-C XL data, cross-linked cysteines displayed a much wider range of pKa than cross-linked lysines (Supplementary Fig. 16), suggesting their varied chemical environments in protein structures. Notably, previous studies have reported similarly extended distances for cysteine−cysteine cross-links[16,31]. Furthermore, the faster kinetics and higher aqueous solubility of SIA compared to SIAB may facilitate the capture of transient or flexible protein conformations, thereby increasing the likelihood of observing longer cross-link distances. Such behavior is consistent with findings that apparent distance violations often reflect structural heterogeneity or protein dynamics rather than technical artifacts[54–56]. Moreover, even homo-bifunctional lysine−lysine linkers such as DSS and DOPA could yield low distance satisfaction rates ranging from 55% to 79% for standard proteins[57]. Furthermore, extended distance (25 Å) has been used as the upper bound for zero-length cross-linkers such as EDC[56] and DMTMM for structural analysis of proteins[55]. These findings, along with molecular dynamics simulations showing conformational flexibility not captured by static models[58], support the view that linker spacer length alone does not dictate cross-linking behavior. Finally, heterobifunctional K−C linkers introduce additional asymmetry and complexity not accounted for in most structural validation tools, underscoring the need for more nuanced approaches in interpreting cross-linking data. Taken together, our results demonstrate that cross-link distance satisfaction is influenced by multiple interrelated factors—including chemical reactivity, protein dynamics, and linker asymmetry—rather than being solely defined by spacer arm length.

Current XL-MS studies have mainly relied on existing high-resolution structures to evaluate cross-links. Here, we have demonstrated that AF-based structural modeling was able to provide additional coverage beyond experimental structures and utilize identified cross-links to validate a large fraction of monomeric models. In addition, AF-based predictions have enabled the assessment of cross-links in multimeric models, facilitating the interpretation of XL-MS data. While the satisfaction rates for inter-protein cross-links were lower than those for intra-protein cross-links, their satisfaction rates improved with increased identification frequency, suggesting that cross-linking reactions are sensitive to structural variability and dynamic conformational states (Supplementary Fig. 9e-h). As cross-link data describes an average ensemble of protein conformations in solution, highly reproducible data would be better suited for characterizing stable structural states. Secondary structural analysis has suggested that the out-of-the-distance cross-links are more likely localized at loop regions, implicating the structural flexibility and dynamics of the identified PPIs. As the existing structures and

predicted models are static, they describe interactions within protein complexes that are specific to a conformational state. However, XL-MS analysis reveals protein interactions among an ensemble of conformational states of protein complexes. Thus, conformation-specific cross-links can only be mapped well onto their corresponding structures[32]. We also find that the K-C cross-linkers are complementary to the commonly used K-K cross-linkers (e.g. DSBSO) in capturing interaction contacts and PPIs, uncovering previously unknown interactions and confirming structural models for known interactions without experimental structures. Moreover, we have shown that combining K-C and K-K cross-links is beneficial in facilitating the modeling of high-order assemblies, such as eEF1 and SERBP1-ribosome complexes. Interestingly, K-C XL-MS analysis has allowed us to identify cross-links that fit better with some AF-models than K-K cross-links, detecting alternative protein conformations. Such conformation-specific K-C cross-links can be used for modeling protein dynamics[10] in future studies. Our results also highlight the advantage of integrating AF-structure prediction with cross-linking data, to validate structural models. In summary, this study has demonstrated the benefits of K-C cross-linking chemistry and complementarity to existing reagents for global profiling of endogenous PPIs from intact cells. AF-based modeling has been effectively integrated to facilitate the interpretation of XL-data and structural analysis of protein complexes that constitute the human interactome. The strategies presented here can be generalized to further advance XL-MS technologies for detailed elucidation of cellular networks from various sample origins to understand PPI-dependent function and regulation of cells.

## Methods

### Materials and reagents

General chemicals were purchased from Fisher Scientific or VWR International. Bovine serum albumin (≥96% purity) was purchased from Sigma-Aldrich. Ac-LR9 peptide (Ac- LADVCAHER, 98% purity) and Ac-SR8 peptide (Ac-SAKAYEHR) were custom ordered from Biomatik (Wilmington, DE). SIA (succinimidyl iodoacetate), SIAB (succinimidyl (4-iodoacetyl)aminobenzoate), and SBAP (succinimidyl 3-(bromoacetamido)propionate) were purchased from Thermo Fisher Scientific. Due to their light sensitivity, these chemicals were stored and handled in the dark during the cross-linking reactions.

### Preparation of cross-linked peptides and proteins

Synthetic peptides Ac-LR9 (Ac-LADVCAHER) and Ac-SR8 (Ac-SAKAYEHR) were used for lysine to cysteine cross-linking with SIA, SIAB and SBAP, respectively. Briefly, 1 mM peptide mixture (Ac-LR9 and Ac-SR8) was cross-linked with each linker in a 1:1 molar ratio of peptide to cross-linker in DMSO. The resulting products were diluted to 10 pmol/μL in 3% ACN/2% formic acid prior to LC MS[n] analysis[27] (Supplementary Methods).

Standard protein BSA and ALDOA were utilized for initial method development. 50 μM BSA or ALDOA in PBS buffer (pH 7.4) was reacted with each linker (SIA, SIAB or SBAP) in a molar ratio of 1:25 for 30 min at room temperature in dark. After quenching by 50 mM ABC and 2 mM Cys for 15 min, cross-linked protein samples were digested prior to LC MS[n] analysis as previously described[27] (Supplementary Methods).

### Oxidation of Cross-linked Peptides

SIA, SIAB and SBAP cross-linked peptides were vacuum dried and dissolved in 100 μL of citrate buffer (50 mM, pH 5.5). Then Chloramine-T was added to the final concentration of 1 mM. The oxidation reaction was carried out for 1 min at room temperature, then quenched with the addition of 10 mM sodium metabisulfite. The oxidized peptides were desalted for LC-MS[n] analysis or further fractionation.

### In vivo cross-linking of HEK 293 cells

HEK 293 cells were harvested and washed with PBS before resuspended in PBS containing 2 mM SIA, 1 mM SIAB, or 2 mM SBAP. The cells were cross-linked for 30 minutes with rotation at 37 °C (or room temperature for SIAB) in the dark. After washing with PBS, the cross-linked cells were lysed using a two-step protein extraction method. The proteins were digested and the cross-linked peptides were separated via SEC and oxidized before further fractionation (Supplementary Methods).

### Liquid chromatography-multistage mass spectrometry (LC-MS[n]) analysis

Cross-linked peptides were analyzed by LC-MS[n] utilizing a Dionex UltiMate™ 3000 (Thermo Fisher, San Jose, CA) coupled on-line to an Orbitrap Fusion™ mass spectrometer (Thermo Fisher, San Jose, CA)[59]. LC-MS[n] data extraction and database searching for the identification of cross-linked peptides were performed similarly as previously described[27,60] (Supplementary Methods).

### Liquid chromatography-tandem mass spectrometry (LC-MS/MS) analysis

The cross-linked peptides were also analyzed by step-HCD MS[2] similar to previously described[33]. Samples were loaded onto a 50 cm×75 μm Acclaim PepMap C18 column and separated over a 120 min gradient of 4% to 25% acetonitrile at a flow rate of 300 nL/min. The step-HCD MS[2] acquisition method was used for the identification of cross-linked peptides. Ions with a charge of 4+ to 8+ in the MS[1] scan were selected for MS[2] analysis. HCD MS[2] scans had a normalized collision energy of 27 with a step of ±6.

### PPI network mapping and analysis

XL-PPI networks were generated from the pair-wise interactions determined by in vivo cross-linking. These interactions were visualized using Gephi v0.10.1 (http://gephi.org). The latest known *Homo sapiens* PPI database on BioGRID (https://thebiogrid.org/), STRING, and BioPlex (http://bioplex.hms.harvard.edu/) were used for data comparison. CORUM (http://mips.helmholtz-muenchen.de/corum) was used to determine the annotated protein complexes that were present in our datasets based on the list of the cross-linked proteins identified here. Functional enrichment was performed using the Gene Ontology Consortium (http://www.geneontology.org) or the R package "ClusterProfiler". Only high confidence annotations for GO cellular compartment and biological process were reported.

### Distance mapping and alphafold prediction for intra-protein cross-links

For each protein, we identified all associated structures in the PDB database by querying the UniProt Database. Structures were filtered if the numbering of the amino acids did not match the sequence. For each cross-link, we computed the Cα-Cα distance of the cross-linked amino acids in each associated structure and recorded the minimal measured distance. If some of the associated structures had multiple copies of the protein (homomers), the distances between the cross-linked amino acids in separate protein chains were measured, and again the minimal distance was recorded. For each protein, we have also generated a single AlphaFold structure model based on two copies of the protein sequence by applying the AlphaFold-Multimer (AFMv2.3, model 1) with default parameters and without templates. For each cross-link, we considered the minimal distances between cross-linked amino acids in the same chain (intra-chain) and the AFM model of two chains (inter-chain).

## Distance mapping and AlphaFold prediction for inter-protein cross-links

We queried the UniProt Database separately for each of the two proteins to identify all associated PDB structures and by intersection of the two lists could identify structures that contained the two cross-linked proteins. Additionally, we generated a structural model of the interaction by applying AFM on the pair of sequences (AFMv2.3, model 1). Furthermore, for each pair of proteins, we queried the STRING database[61] for an interaction score. Homologous structures were found by applying BLAST on each protein separately and finding all PDB structures with chains containing the sequence with at least 70% coverage and 70% sequence identity. A pair of cross-linked proteins were considered to have a homolog if they had a PDB structure that contained a homologous chain for each of them.

## Integrative structure modeling

For the integrative assembly of large complexes, we used CombFold[62] default pipeline, using AFM to compute different subcomplexes structures of sizes two to five subunits and assembled them using the CombFold algorithm. We supplied the cross-links as distance restraints when running CombFold. SERBP1 was modeled using AlphaFold3. ChimeraX[63] was used for structures visualization.

## Secondary structure analysis

We used DSSP[64] to determine the secondary structure for each pair of cross-linked residues with Cα-Cα distances below 30 Å. We categorized the secondary structures into three main groups: helices, sheets, and loops. To compute background secondary structure distribution of lysine and cysteine, we used DSSP to compute the secondary structure of all the AlphaFold models in our datasets. For each dataset (intra-protein and inter-protein K-C cross-links, intra-protein and inter-protein DBSO cross-links), we aggregated the counts of the secondary structure types for the lysine and cysteine positions and normalized these counts using the background secondary structure counts obtained from the AlphaFold analysis.

## Reporting summary

Further information on research design is available in the Nature Portfolio Reporting Summary linked to this article.

## Data availability

Unless otherwise stated, all data supporting the results of this study can be found in the article, supplementary, and source data files. The mass spectrometry proteomics data have been deposited to the ProteomeXchange Consortium via the PRIDE[65] partner repository with the dataset identifier PXD055169. All other data needed to evaluate the conclusions in the paper are present in the paper and/or supporting information. In addition, the following published structures were used for cross-link mapping or structural modeling: 4F5S, 5T46, 6XMH, 7FGM, 7PU5, 7W37, 7WU7, 7NVL, 8EW2. The models are available in ModelArchive (www.modelarchive.org) with the accession code ma-hc4kr. Source data are provided with this paper.

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

## Acknowledgements

The authors thank Drs. A.L. Burlingame, Peter Baker, Robert Chalkley and Michael Trnka for the support of Protein Prospector. This work was supported by National Institutes of Health grants R35GM145249 to L.H.

## Author contributions

L.H. conceived the study and directed the research. F.J., L.H., M.B. and D.S. designed experiments. F.J. carried out XL-MS experiments and data analysis. C.Y. performed initial cysteine oxidation experiments and assisted with data acquisition and analysis. M.B. and D.S. performed AF-based structural analysis and integrative modeling. B.S. assisted on structural analysis. B.E.W. assisted on the optimization of cysteine oxidation. L.H., D.S., F.J., M.B., C.Y. and B.S. contributed to the writing of the manuscript.

## Competing interests

The authors declare no competing interests.
