## [Transparent Peer Review file · Nature Communications]

Cysteine-enabled Cleavability to Advance Cross-linking Mass Spectrometry for Global Analysis of Endogenous Protein-Protein Interactions

Corresponding Author: Dr Lan Huang

Version 0:

Reviewer comments:

Reviewer #1

(Remarks to the Author)

****Summary****

The authors describe a novel workflow for utilizing the heterobifunctional nature of three K-S crosslinkers: SIA (succinimidyl iodoacetamide, 1.5 Å), SIAB (succinimidyl (4-iodoacetyl)aminobenzoate, 10.6 Å) and SBAP (succinimidyl 3-(bromoacetamido)propionate, 6.2 Å). The authors demonstrate the utility of converting these commercially available crosslinkers into MS-cleavable products to boost XL-MS identifications for derived PPI's. They then describe the orthogonality of these crosslinkers to the more common NHS ester hydrolysis K-K crosslinkers, to identify unique PPI's that are not accessible otherwise, in the context of in-cell crosslinking. Following this, the authors perform AlphaFold based structural validation of intra-protein crosslinks, as well as AlphaFold multimer based structural validation of homo- and hetero-multimeric protein complexes. Importantly, the authors demonstrate the enriched utility of K-K vs K-C crosslinkers in probing different secondary structures for protein structure analysis.

This work represents an important contribution to the field of cross-linking MS, in the context of in-cell crosslinking and the leveraging of these crosslinks against protein structure prediction algorithms.

****Major Corrections****

- I may have missed something here but, there is no mention of biological replicates pertaining to the in-cell crosslinking work. Generally, one would expect some statistical criteria to deal with the high-throughput and relatively high rate of false crosslinks (e.g. crosslink present in at least 2 of 3 replicates). However, I understand that this work is designed to serve as a framework for XL-MS generation and incorporation into multimer modelling. If this work is all done on one replicate, I would suggest this being mentioned clearly in the Methods section. Further, at a minimum, an MS2 CSM score cut-off threshold should be applied to the data to reduce false discoveries.

****Minor Corrections****

- Figure 1b: Following the crosslinking step, the 'making cysteine cleavable' step is described by the oxidation to generate a S=O bond, but the molecular structure is the same as before (without the sulfonyl group). Please correct this.
- Supplementary Figure 7: It would be useful to explain what pLDDT is, as many readers would not know its importance as a residue specific confidence metric output from AlphaFold.
- Whilst I am confident that the pre-MS oxidation of cysteines to generate MS-cleavable crosslinks would improve crosslink identifications; a comparison of crosslink-identifications for standard crosslinks vs oxidised crosslinks would be useful. Could the authors comment on this.
- Could the authors comment on the potential for side-reactions when exposing the peptides to the chloramine-T oxidising agent? Is there the possibility that there would be additional oxidation events on other amino acids from the reaction step,

which would in turn modify amino acid masses and impact data processing? A quick open modification search of the RAW data would shed light on this.

The authors describe that "The oxidation reaction was carried out for 1 min at room temperature" - where other time points evaluated?

The authors state "For those only identified with cross-linked lysines, 167 have not been reported in our previous *_in vivo_* K-K XL-data" - Can the authors expand on why they think if there is anything special about these proteins?

Reviewer #2

(Remarks to the Author)

Reviewer #3

(Remarks to the Author)

Reviewer #4

(Remarks to the Author)

Jiao and colleagues present an interesting workflow for enabling MS-cleavability of three commercially available lysine-to-cysteine cross-linkers with different spacer arm lengths by oxidation, which exhibits a benefit for the application of such heterobifunctional cross-linkers for both in-vitro and in-vivo proteome-wide interactomics studies. Traditionally, those complex studies were only possible using conventional lysine-to-lysine cross-linkers, limiting the information gained to lysine-containing interaction interfaces. The authors have applied different chemical and computational strategies such as AlphaFold2 and secondary structure prediction to collect evidence for the validity of their cross-link identifications. Moreover, they suggest a benefit of the induced cleavability by oxidation for cysteine modification mapping in chemical proteomics. With few minor exceptions, the manuscript is written clearly and comprehensively. Work in the field of heterobifunctional cross-linkers is critically important as certain proteins of interest fully lack lysines or lack them in regions of interest. The authors put a tremendous effort in validating their identifications using sophisticated state-of-the-art tools and methodologies, which are advancing to standard workflows in the XL-MS field. However, the combination of AF and XL-MS has been shown extensively for conventional K-K cross-linkers, which brings down the novelty of this study to some extent. I appreciate the comprehensive benchmarking against K-K cross-linkers, highlighting e. g. that K-C and K-K cross-linkers align well in terms of proteomic / organelle coverage (Suppl. Fig. 6). However, I would like the authors to address my following concerns.

Major concerns:

1. p. 8: The authors identified > 150 unique K-C cross-links in purified BSA and argue that the agreement of 3 different cross-linking reagents (Fig. 2i) suggests robustness of the approach. This may be true for a more complex system but when only one protein is being used for the database search, the false positive space is much smaller. The authors must clarify how the BSA samples were searched. I suggest to also add to Fig. 2 a panel with density plots after structure mapping showing the fraction of cross-links within the distance cutoff vs a randomly sampled K-C distance distribution. If the majority of identified cross-links does not exceed this cutoff, the identifications are more convincing.
2. On the same line, a general 30 Å distance cutoff for all three cross-linking reagents does not seem fair as they vary drastically in length. Considering the length of a K side chain (~ 7.6 Å) and the length of a C side chain (~ 3.4 Å), one can't allow more flexibility than 15 Å for SIA or 20 Å for SBAP. This will have large effect on the "limited overlap at the residue-to-residue level" (p. 19, line 413).
3. p. 8 & p.9: When the authors state the number of identified cross-links from BSA and HEK293, they must report in the main text at which FDR, on which level (CSM, XL, PPIs) and with which strategy (inter/intra separate, context-sensitive, etc.) the data was filtered. This information is missing in both manuscript and suppl. information, which strongly hinders reproducibility. The search engine ProteinProspector must be mentioned in the main text.
4. p. 10: Max. 76% intra-link satisfaction on PDB structures is suspiciously low, especially with such relaxed distance cutoffs. Bartolec et al. PNAS 2023 have reported a > 89% satisfaction rate for intra-links (only DMTMM inter-chain links are below 80%). Also, the drastically low satisfaction of inter-links on PDB structures does not align with previous reports. On p. 16, line 345 resulting prediction match to 40% of cross-links which does not support the reliability of either the cross-link IDs or the AF prediction.
5. p. 13, line 274: Claiming that 59.1% of PPIs are new because they haven't been reported previously and don't match to any public repositories does not intrinsically indicate their novelty. This claim should be removed as those entries can also be interpreted as contradictory to prior studies, arising from potentially false-positive identifications.

Minor remarks:

1. p. 3, line 31: What is an "enabling technology"?
2. Cysteine residues are overall less abundant than lysine residues (~3% vs ~6%), making them a less attractive target for cross-linking. Can the K-C cross-linkers modify both oxidized C which form disulfide bridges and the reduced thiols? If one

is limited to thiols, there are even less targets available.

3. p. 9, line 168: Using stepped-HCD MS2, you identified 57059 K-C peptides, ~15% less compared to MS2-MS3 acquisition. This opposes other studies, e.g. <https://doi.org/10.1021/acs.analchem.1c04812> (Suppl. Fig. 1), <https://doi.org/10.1038/s41467-022-31701-w> (Fig. 9), <https://doi.org/10.1021/acs.jproteome.8b00947> (Fig. 6). Can you explain this divergence?

4. Can you show that other search engines such as MSAnnika or XlinkX align with the data obtained from ProteinProspector?

5. p. 9, line 182: 847 proteins were found to be cross-linked only by K-C cross-linkers. Why is that? What are the unique features of those proteins? Is the number of Cs higher on those proteins compared to K-K cross-linked proteins?

6. p. 9, line 183: I don't understand this statement and corresponding Suppl. Fig. 5c, could you please rephrase it? Why can "K-C XL-proteomes contain proteins identified with lysine- and/or cysteine-containing peptides"? The "or" implies that it does not have to be K-C but also K-K or C-C when using those cross-linkers.

7. When you're comparing your K-C XL proteome to K-K XL proteomes you have reported earlier, you need to emphasize that DSBSO has a much higher cross-linkable distance than all 3 K-C cross-linkers. Please include a statement that this reduces fair comparability between the K-C and K-K datasets.

8. p. 12, line 236: The relationship between AlphaFold quality metrics and cross-link satisfaction was reported in numerous works that should be cited. For example: <https://doi.org/10.15252/msb.202311544>, <https://doi.org/10.1101/2023.12.03.569778>, <https://doi.org/10.1101/2023.07.19.549678>

9. p. 13, line 266: As you're correctly stating that predicting protein fragments may improve satisfaction rates, why haven't you tried that for selected violating cross-links?

10. p. 13, line 271: You created a PPI network with 2007 nodes (corresponding to crosslinked proteins) and 6589 edges (PPIs?). Could you please change the terms nodes and edges to their biological meanings – also in Suppl. Fig. 8a? I could not find what 6589 refers to in the main text.

11. Suppl. Fig. 8b should be separated into several venn diagrams showing the overlap of the K-C data with other data or databases. Especially the overlap with the BioPlex data is extremely low in my experience.

12. Suppl. Fig. 8c and p. 13, line 276 suggest that the majority of PPIs could not be mapped to the STRING network. Have you filtered the STRING database for certain evidences or did you consider all entries in the database? If you considered all entries, the overlap with STRING should be higher than ~10% (<https://doi.org/10.1038/s41592-024-02478-1>, Fig. 6b).

13. Fig. 3d needs fully labeled axis. Instead of using ProteomicsDB I suggest to search the RAW data again with MaxQuant and calculate an iBAQ specifically for the samples used in this study. A plot as in <https://doi.org/10.1021/acs.analchem.3c04682> Fig. 2 would strengthen the claim that cross-links were identified throughout the whole proteome.

14. p. 17, line 356: 12% of loops in eEF1G is objectively not a "high percentage", please rephrase.

15. p. 19, line 400: You mention more than once the benefit of Cys oxidation induced cleavability for chemical proteomics. Please include in the discussion a more detailed hypothesis of what can be gained from applying this method?

16. p. 20, line 432: You argue that SIAB cross-links are more in agreement with the distance threshold due to its lesser rigidity. Could you elaborate whether the rigidity of the cross-linker still plays a role when it is bound to a flexible lysine? Moreover, SIA is 7x shorter than SIAB, which intuitively should have the highest effect on whether a cross-link stays within a 30 Å cutoff.

17. On the same line: According to Suppl. Fig. 11, K-K cross-links are more prone to fall into loop regions, potentially yielding more overlength links when mapped on predicted structures as AF might fail to position those regions correctly. K-C linkages fall predominantly into Helices and Sheets, which usually have higher pLDDT scores. Would you not think that this should be reflected in higher cross-link satisfaction rates?

18. p. 22, line 474: Was the mixture "reacted with each linker...in dark" because the cross-linkers are light sensitive? In the ThermoFisher User Guide for SIA (https://assets.thermofisher.com/TFS-Assets/LSG/manuals/MAN0011372_SIA_UG.pdf) it is not mentioned to incubate the linker in the dark.

19. p. 24, line 519: "Structures were filtered out if numbering of the aa did not match the sequence." Does that mean that no structure mapping was performed the numbering of the PDB was off by even 1 position from the numbering of your cross-linking result? This would remove a lot of PDB structures from the pool of accepted structures because the numbering of PDB structures is seldomly in accordance with the numbering of a FASTA retrieved from UniProt. The authors should find means to consider mapping links to all PDB structures and fix the numbering by calculating a protein-specific offset if needed.

20. Fig. 4d needs a y-axis label. "AlphaFold confidence" can be said in the title, but the y-axis should contain the actual metric (PAE).

21. Fig. 6: add a density plot visualizing cross-link satisfaction.

22. The RAW files uploaded to PRIDE may be of general interest as the data availability for heterobifunctional cross-linkers is sparse, thus it might be useful to upload an assignment table for RAW files to experiment and/or FASTA file and/or figure they were used for.

23. The XL-MS community puts a strong focus on uniform data sharing, so please consider uploading your data to PRIDE in the newest mzIdentML format.

24. The title could emphasize more the heterobifunctional character of the cleavable cross-linker.

25. For density plots, consider showing the median, too.

Version 1:

Reviewer comments:

Reviewer #4

(Remarks to the Author)

The authors have done a good job addressing the reviewers' comments, and I am satisfied with the revised manuscript overall. However, one major concern remains unresolved, which is my initial Question 3 regarding the fidelity of the search results.

Major Concern A3: The authors state that their MSⁿ-based strategy yielded no decoy identifications and thus no FDR estimation, which is highly unusual. Their search was conducted against the entire *Bos taurus* (6,006 entries) and *Homo sapiens* (20,387 entries) proteomes. The MSⁿ-based strategy searches the two linked peptides individually using the precursors obtained in MS2 and fragment ions in subsequent MS3. To the best of my knowledge, all peptide searches generate decoy hits and FDR is calculated based on number of target and decoy identifications. Could it be that ProteinProspector simply does not export them so the authors do not see any decoy hits? Regardless, FDR estimation (even for MS²–MS³) is essential for validation, and this must be performed and reported; without it, the results remain questionable. This concern also relates to Minor A4, where I suggested cross-validation using alternative software (e.g., MSAnnika, XlinkX), given that ProteinProspector is not widely adopted in this field. While the authors note that MSAnnika and XlinkX do not support asymmetric crosslinkers, xiSearch offers better flexibility. Could the search be done with xiSearch to strengthen the findings?

Minor Comment A18: Please include a brief explanation in the Methods section to highlight the reagent's instability, ensuring readers are aware of this limitation.

Reviewer #5

(Remarks to the Author)

Substitute Reviewer #1

I was invited to review the answers to the comments raised by my colleague. Overall, paper is good, novel, has beautiful in-cell cross-linking in human cells, answers are satisfactory and only few clarifications/changes remain. However, I must note that I am a structural biologist, expert in cryo-EM, with a lot of knowledge in mass spectrometry, and I can understand the reviewer's comments and answers; however, I cannot comment on other technical parts of this manuscript. Below you see my comments:

Comment #1: Question #1, Answer to Question #1.

The authors have improved the manuscript by clearly indicating their replicates. However, they must improve Table S6, with all data regarding the exact cross-links covering those complexes. It is extremely challenging now to search and map those on protein complexes for this reviewer, as well as seeing how many, or which cross-links were captured per reported CORUM complex.

Comment #2: Question #2, Answer to Question #2.

Properly addressed.

Comment #3: Question #3, Answer to Question #3.

Properly addressed, I guess it is now Figure S9. However: The authors must understand and revise the presentation of AlphaFold models in the text, versus PDB models: AlphaFold models are predictions, whereas PDB models are experimental structures. Cross-links cannot be validated by a prediction, but a prediction is validated by the cross-linking data.

Comment #4: Question #4, Answer to Question #4.

Properly addressed, along with the limitations and challenges; Figure 1R is on the spot.

Comment #5: Question #5, Answer to Question #5.

Properly addressed. The added revised text is addressing the reviewer's concern.

Comment #6: Question #6, Answer to Question #6.

The authors did not respond to this comment. They must clarify if they benchmarked cross-linking time or not.

Comment #7: Question #7, Answer to Question #7.

Properly addressed.

Version 2:

Reviewer comments:

Reviewer #4

(Remarks to the Author)

While I still had some reservations about the FDR control method in the ProteinProspector pipeline, in this round of the revision the authors compared their results with an independent search engine, MeroX. The two pipelines showed strong agreement (85% overlap) in identification. MeroX reports an entrapment-based false-positive rate of only ~2% at a 1% FDR, while ProteinProspector identified zero false positives at any FDR threshold. I think this information should be included in the manuscript, which also emphasizes their pipeline is somewhat unique and robust. Then I would be happy to accept its publication.

Detailed responses to reviewers' comments:

Note: Figures included solely to address reviewers' comments are labeled as **Figure #R** (# represents figure number for responses). Figures that appear in the manuscript retain the same figure number and description here as in the main text.

Reviewer #1

Q1: I may have missed something here but, there is no mention of biological replicates pertaining to the in-cell crosslinking work. Generally, one would expect some statistical criteria to deal with the high-throughput and relatively high rate of false crosslinks (e.g. crosslink present in at least 2 of 3 replicates). However, I understand that this work is designed to serve as a framework for XL-MS generation and incorporation into multimer modelling. If this work is all done on one replicate, I would suggest this being mentioned clearly in the Methods section. Further, at a minimum, an MS2 CSM score cut-off threshold should be applied to the data to reduce false discoveries.

A1: We apologize for any confusion and would like to clarify that we have performed at least three biological replicates for in-cell cross-linking using each linker. Specifically, we conducted four biological replicates for SIA, six replicates for SIAB, and three replicates for SBAP. All replicates have been analyzed by LC MSⁿ. Because our study primarily focuses on establishing an MSⁿ-based XL-MS workflow for identification of K-C cross-links, fewer replicates were selected for MS²-based analysis (two SIA replicates, three SIAB replicates, and three SBAP replicates). In comparison to MSⁿ-, MS²-based analysis has proven to be extremely challenging and time consuming, as existing XL software tools are not well-suited to identify heterobifunctional K-C cross-links at the proteome-level. In this work, 1% FDR cutoff at the CSM level was applied for all MS²-based identification. The details on biological replicates and FDRs have been added to the main text and experimental methods for clarity.

Q2: Figure 1b: Following the crosslinking step, the 'making cysteine cleavable' step is described by the oxidation to generate a S=O bond, but the molecular structure is the same as before (without the sulfonyl group). Please correct this.

A2: We apologize for the oversight and have corrected the schematic in Figure 1b.

Q3: Supplementary Figure 7: It would be useful to explain what pLDDT is, as many readers would not know its importance as a residue specific confidence metric output from AlphaFold.

A3: We thank the reviewer for the suggestion and have added the detailed description of pLDDT as the "predicted local distance difference test" in the text and abbreviation section.

Q4: Whilst I am confident that the pre-MS oxidation of cysteines to generate MS-cleavable crosslinks would improve crosslink identifications; a comparison of crosslink-identifications

for standard crosslinks vs oxidised crosslinks would be useful. Could the authors comment on this.

A4: We thank the reviewer for the suggestion. However, it is difficult to thoroughly compare the identification of standard (non-cleavable, unoxidized) and oxidized (cleavable) cross-links using the heterobifunctional cross-linkers described in this work due to several reasons. First, almost all existing MS²-based cross-link software packages are designed for homobifunctional cross-linkers, not suited for searching heterobifunctional cross-link data. Secondly, very few software can fairly assess MS²-based identification for both cleavable and non-cleavable cross-links. In addition, since the MS-cleavable site is not within the linker spacer region but later introduced at the side chain of the cross-linked cysteine, this yields an asymmetric cleavable structure between the cross-linked residues. CID-induced cleavage leads to unequal cross-linker fragments retained at the cross-linked lysine and cysteine respectively, further complicating MS²-based searches. This unique feature requires XL software to allow us not only to set heterobifunctional cleavable reagents for searching, but also to specify crosslinker fragments associated with specific residues with defined formula and masses. As a result, we have relied on Protein Prospector (<https://prospector.ucsf.edu/prospector/mshome.htm>) for searching MS² data, as it is capable of handling both noncleavable and cleavable cross-linkers, but also allowing database searching against larger protein databases using a mass modification-based approach (Chu F, et al, *Mol Cell Proteomics*. 2010). Importantly, Protein Prospector is capable of identifying heterobifunctional MS-cleavable cross-links with asymmetric cleavages by allowing us to define specific modifications at cross-linked cysteines and lysines after cross-link cleavage. Unfortunately, such database searching is extremely slow, taking more than a month to complete the analysis of one set of MS² data against a restrictive protein database composed of 3,359 proteins that were determined from our SIA MS³ searches in combination with our published DSSO and DBrASO XL-MS data (Jiao, F. et al, *Anal. Chem.* 2020; *Anal. Chem.* 2023). Therefore, we were not able to perform extensive MS² analysis for the suggested comparison. While MS²-based analysis is more sensitive, MS³-based analysis has proven to be significantly more straightforward, faster, and easier for cross-link identification—especially at the proteome level, regardless of homo- or heterobifunctional cross-linkers used. Thus, the main focus of our work presented here is to establish an MSⁿ-based analysis workflow for accurate identification of K-C cross-links. Nonetheless, to address the reviewer's point, we further evaluated the MS²-based cross-link identifications from a SIA cross-linked sample using Protein Prospector. Conversion of non-cleavable SIA cross-links to cleavable cross-links from a SEC-HpHt fraction increased the total number of unique cross-link spectrum matches (655 vs. 352) at 1% FDR, yielding a higher number of unique cross-linked peptides (367 vs. 257). When comparing cross-linked peptide scores, oxidation-induced cleavability has shifted the score distribution towards higher scores (**Figure 1R**), thus increasing the identification confidence. These observations are consistent with published

results that MS-cleavability generally improves peptide backbone fragmentation and sequence coverage to enhance cross-link identification (Kolbowski, L., et al. *Anal. Chem.*, 2022).

Figure 1R. Score distribution of SIA cross-linked peptides before and after oxidation using MS²-based analysis. A SEC-HpHt fraction of SIA cross-linked HEK293 cells was analyzed by stepped-energy HCD-MS² and the resulting XL-MS data was searched using Protein Prospector. Cross-linked peptides were identified at 1% FDR.

Q5: *Could the authors comment on the potential for side-reactions when exposing the peptides to the chloramine-T oxidizing agent? Is there the possibility that there would be additional oxidation events on other amino acids from the reaction step, which would in turn modify amino acid masses and impact data processing? A quick open modification search of the RAW data would shed light on this.*

A5: We thank the reviewer for the question. Previous reports have indicated that chloramine-T (Ch-T) oxidation is specific to methionine and cysteine, not other amino acids (Shechter, et al, *Biochemistry*, 1975, 14). To evaluate oxidation specificity, we have tested Ch-T oxidation on two standard peptides and a protein digest. LC MS² analysis of carbamidomethylated Ac-LR9 (LADVC(carbam)HER) has revealed that chloramine-T oxidation completely converted the peptide to its oxidized form within 1 min (Supplementary Figure 1). Furthermore, MS² sequencing of the oxidized form determined that the oxidation is specific to the carbamidomethylated cysteine and not any other residues in the sequence, including histidine. Ch-T oxidation was also performed on Ac-SR8 peptide (Ac-SAKYEHR). LC MS² analysis showed that no new products were detected following oxidation, indicating Ch-T did not oxidize any amino acids in Ac-SR8 peptide sequence, including tyrosine (**Figure 2R**). Finally, we analyzed a BSA digest before and Ch-T oxidation. Apart from the oxidation of methionine and carbamidomethylated cysteines, database searching revealed that tryptophan oxidation was

increased following the Ch-T reaction. This is not surprising as tryptophan is prone to oxidation during sample preparation. To investigate whether adding tryptophan oxidation as a variable modification would have a positive impact on K-C cross-link identification, we have re-searched LC MSⁿ data for two sets of SIA in-cell cross-linked samples. When including tryptophan oxidation as a variable modification, only a marginal (~5%) increase in cross-link identifications were observed (e.g. 2,014 vs. 2,124). Since tryptophan is one of the least abundant amino acids in human protein sequences (~1%), we decided not to re-search all XL-MS datasets to include tryptophan oxidation, but mentioned the observation in the discussion as follows on page 23:

“Although chloramine-T induced oxidation is highly selective for carbamidomethylated cysteines under our experimental conditions, analysis of BSA XL-data suggested an enhanced oxidation of tryptophan. Including tryptophan oxidation as a variable modification can increase peptide identification; however, its overall impact remains limited given the low abundance of tryptophan residues in the human proteome (~1%). Nonetheless, considering tryptophan oxidation during database searching could be beneficial particularly when analyzing proteins rich in tryptophan or peptides of interest that contain tryptophan residues.”

Figure 2R. LC MS² analysis of the standard peptide SR-8 (Ac-SAKAYHER) before and after chloramine-T oxidation at 1 min.

Q6: *The authors describe that "The oxidation reaction was carried out for 1 min at room temperature" - where other time points evaluated?*

A6: We thank the reviewer for the question. Based on our peptide analysis, the oxidation reaction was observed near completion within 1 min (Supplementary Figure 1), and there was no obvious difference in yield when using longer times up to 5 mins. Therefore, all experiments were performed using a reaction time of 1 min.

Q7: *The authors state "For those only identified with cross-linked lysines, 167 have not been reported in our previous in vivo K-K XL-data" - Can the authors expand on why they think if there is anything special about these proteins?*

A7: We thank the reviewer for pointing this out. Upon re-examining our data, we identified an

error in the number of proteins with the stated features in the original submission. Instead of 167, only 82 proteins were identified as cross-linked proteins with only lysine-containing peptides and were not identified by our previous *in vivo* K-K XL-data. We apologize for this oversight and have revised the text accordingly. We believe these observations to be interesting as K-C cross-linkers enable the identification of interaction partners through lysine-containing peptides, cysteine-containing peptides, or a combination of both. While many of these proteins have lysines at their interaction interfaces, their interactions with specific binding partners were only identified through K-C cross-links. We suspect that the identification of these PPIs unique to K-C linkers is most likely due to the availability and accessibility of lysines-cysteine pairs at the interaction interfaces and the detectability of the resulting cross-linked peptides. To test this, we examined protein primary sequences and structures of two selected interactions. These two examples have been included in the main text as follows on page 11-12:

“To understand why these proteins are unique to K-C linkers, we examined two selected interactions in more details. For example, the interaction between EIF4E and EIF4G1 was identified through the cross-link of EIF4E:K192 to EIF4G1:C662, where EIF4E was detected with only cross-linked lysine-containing peptides. This cross-link was reproducibly identified by all three K-C linkers, supporting its validity. Notably, although eight lysines are distributed within residues 600–720 of EIF4G1, none were found to be cross-linked to EIF4E in previous XL-MS studies using lysine-reactive homobifunctional linkers (Liu, F., et.al., *Nat Commun*, 2018; Kuma, Y., et.al., *Mol & Cell Proteomics*, 2020; Wheat, A., et. al., *PNAS*, 2021; Jiao, F., et.al., *Anal Chem*, 2021; Jiao, F., et.al., *J. Proteome Res*, 2024). Analysis of the existing (PDB: 5T46) and AlphaFold3 predicted structures of the human EIF4E-EIF4G complex revealed that six of these lysines in EIF4G1 are located 30~40Å from K192 of EIF4E, while the remaining two lysines are much farther away (85.1 Å and 86.6 Å) (Supplementary Figure 8a,b). Interestingly, this EIF4E-EIF4G1 interaction was captured exclusively using K-C cross-linking, made possible by the proximity of EIF4G1:C662 to EIF4E:K192 (10.3 Å).

Another example involves the interaction between S100A10 and ANXA2 through the identification of S100A10:K47-ANXA2:C9 cross-link. Based on the AlphaFold3 predicted structure of the S100A10-ANXA2 complex, both K10 (11.4 Å) and K28 (21.2 Å) of ANXA2 are close to K47 of S100A10 and thus their cross-links should be observed using existing lysine cross-linkers (Supplementary Fig. 8c). Despite their proximity, ANXA2:K10 and ANXA2:K28 have not been found cross-linked to S100A10:K47 in previous studies (Liu, F., et.al., *Nat Commun*, 2018; Kuma, Y., et.al., *Mol & Cell Proteomics*, 2020; Wheat, A., et. al., *PNAS*, 2021; Jiao, F., et.al., *Anal Chem*, 2021; Jiao, F., et.al., *J. Proteome Res*, 2024). This is most likely due to the detectability of the resulting tryptic peptides. In this work, cross-linking of ANXA2:C9 to S100A10:K47 involved the identification of a 10-residue long tryptic peptide (MSTVHEILCK) of ANXA2. Formation of an ANXA2:K10 or ANXA2:K28 cross-link to S100A10:K47 cross-link using lysine-specific cross-linkers would both result in much longer

(28-residue) peptides and potentially impeding cross-link detectability. These examples suggest that the exclusive capture of these proteins' interactions by K–C linkers is most likely attributed to the availability and accessibility of lysine-cysteine pairs at interaction interfaces, as well as the detectability of resulting cross-linked peptides.”

Supplemental Fig. 8. **Structural mapping of the identified K-C cross-links and potential K-K cross-links from the two selected interactions.** **a.** Structural mapping of the EIF4E-EIF4G complex with partial sequences (PDB ID: 5T46, EIF4E:31-206, EIF4G1:608-642) to illustrate the distances between EIF4E:K192 and EIF4G1:K609, K610, and K629. **b.** Analysis of the AF-predicted structure of the EIF4E-EIF4G1:C662 complex to map the identified EIF4E:K192-EIF4G1:C662 cross-link (shown in red) and determine the distances between EIF4E:K192 and the eight lysines of nearby C662 of EIF4G1 (shown in blue). **c.** Analysis of the AF-predicted structure of the S100A10-ANXA2 complex to map the identified K-C cross-link (i.e. S100A10:K47-ANXA2:C9) (shown in red) and illustrate the proximity between S100A10:K47 and the lysine residues (i.e. K10, K28) nearby C9 of ANXA2 (shown in blue).

Reviewers #2, #3, and #4

Major concerns:

Q1. p. 8: *The authors identified > 150 unique K-C cross-links in purified BSA and argue that the agreement of 3 different cross-linking reagents (Fig. 2i) suggests robustness of the approach. This may be true for a more complex system but when only one protein is being used for the database search, the false positive space is much smaller. The authors must clarify how the BSA samples were searched. I suggest to also add to Fig. 2 a panel with density plots after structure mapping showing the fraction of cross-links within the distance cutoff vs a randomly sampled K-C distance distribution. If the majority of identified cross-links does not exceed this cutoff, the identifications are more convincing.*

A1: We thank the reviewers for the question and apologize for any confusion. Although the cross-linking reaction was performed on purified BSA, the extracted MS³ data was searched against a concatenated target-decoy database composed of Bos Taurus proteome containing 6,006 protein entries. This detail has been added to the Supplemental information in the section of “Identification of Cross-linked Peptides by MSⁿ analysis”. Based on the reviewers’ suggestion, we plotted the distance distribution of the identified and random K-C cross-links of BSA here (**Figure 3R**), however, the separation between the identified and random cross-links was marginal. To prevent any potential random cross-links detected in our analysis, we have further evaluated K-C cross-linking of BSA under different conditions using SIA, including changes in protein concentrations, linker to protein ratios, reaction times, reducing and quenching conditions, etc. However, regardless of the experimental conditions, the separation between the distance distributions of the identified and random K-C cross-links of BSA was similar, suggesting that such analysis is not effective for validating cross-links in single proteins. This is not surprising as BSA is a relatively small, globular, and well-folded protein, and the number of possible long-distance random cross-link pairs is limited by the protein’s compact structure. Thus, random and true cross-links often fall into overlapping distance ranges for a single protein. Therefore, we decided not to include the random cross-link distribution in the main Figure to prevent confusion. Instead, we have updated Figure 2 with the BSA results derived from the new experiments under the most stringent conditions for all three linkers. With three biological replicates for each linker, a total of 149, 85, and 224 K-C cross-links were identified using SIA, SIAB, and SBAP respectively, which have been updated in Figure 2 and Supplementary Table 1a-c. When mapped to a high resolution BSA structure (PDB:4F5S), the median C α -C α distances are 25.3 Å, 26.0 Å and 26.5 Å for SIA, SIAB and SBAP, respectively (Supplementary Table 1) and their distribution was illustrated in Supplementary Figure 4d.

Figure 3R. Distance distributions of the SIA, SIAB, SBAP cross-links identified in this work when mapped to a high resolution BSA structure (PDB: 4F5S). For comparison, random K-C cross-links of BSA were plotted here.

Q2. On the same line, a general 30 Å distance cutoff for all three cross-linking reagents does not seem fair as they vary drastically in length. Considering the length of a K side chain (~ 7.6 Å) and the length of a C side chain (~ 3.4 Å), one can't allow more flexibility than 15 Å for SIA or 20 Å for SBAP. This will have large effect on the “limited overlap at the residue-to-residue level” (p. 19, line 413).

A2: We thank the reviewers for pointing this out. To examine whether the overlaps at the residue-to-residue level among the three linkers change with cross-link distances, we performed multiple comparisons using BSA K-C cross-links with different Cα-Cα distance thresholds for SIA (15Å, 20Å, 25Å, 30Å) but constant distances for SBAP (30Å) and SIAB (30Å) (Figure 4). The Cα-Cα distances of the cross-links were obtained when mapped to a high resolution BSA structure (PDB:4F5S), and only reproducible cross-links (3 out of 3 biological replicates) were used for comparison. As shown below, when the distance threshold of SIA cross-link decreased (30Å→15Å), the linkage overlaps among the three linkers decreased (10.5%→1.4%) (**Figure 4R**). These results suggest that relaxing the distance threshold for SIA enabled better overlaps with the two longer linkers (SIAB and SBAP), suggesting SIA could capture longer cross-links than expected. When we restrict all cross-links to shorter distances concurrently, their overlaps did not change much. Collectively, these results indicate that cross-link overlaps among the three linkers are not directly related to cross-link distances.

Figure 4R. Effect of distance thresholds on the overlap of SIA cross-links with SIAB and SBAP cross-links in BSA. The distance thresholds for SIA cross-links varied from 15Å to 30Å, while the distance thresholds for SBAP and SIAB cross-links remained constant (30Å). The distance thresholds were indicated for each Venn diagram. The total number of SIA cross-links for comparison decreased when the distance threshold decreased.

We agree with the reviewers that the expected distance thresholds for each K-C linker are expected to correlate with their spacer arm lengths. During the re-examination of the spacer

arm lengths for SIA, SBAP and SIAB, we realized that the initial values obtained from the Thermo website for SIA (1.5 Å) and SBAP (6.2 Å) were incorrectly calculated with two missing bonds (**Figure 5R**). Therefore, we recalculated their distances using Spartan (Wavefunction, Inc.), yielding spacer arm lengths of 4.2 Å for SIA and 9.0 Å for SBAP, while SIAB's spacer arm remained as 10.6 Å. Based on these values, we expect that SBAP could capture cross-links with similar distances to SIAB cross-links, while SIA cross-links would have distances ~ 5 Å shorter. However, the median C α -C α distances for BSA cross-links were determined to be 25.3 Å, 26.5 Å, and 26.0 Å for SIA, SBAP, and SIAB, respectively. Since BSA has many disulfide bonds, disulfide shuffling may contribute to the formation of unexpected cross-links. Thus, we performed XL-MS experiments using another standard protein—rabbit aldolase—that does not carry any disulfide bonds. As a result, the median distances of its cross-links were determined to be 23.4 Å (SIA), 25.2 Å (SBAP), and 26.8 Å (SIAB). These results further suggest that SIA can capture cross-links with distances longer than expected. This observation is most likely attributed to the chemical properties and high reactivity of SIA. In addition, cysteine is highly reactive, and its reactivity is strongly influenced by its microenvironment. Structural evolution analyses have shown that the local environment can significantly affect cysteine's flexibility and dynamic behavior. The higher reactivity and faster kinetics of cysteine cross-linking may enable more effective capture of weak or transient protein interactions. Additionally, cysteine plays an important role in the dynamic regulation of protein structure through disulfide bond formation and rearrangement. Thus, mapping cysteine-based cross-links onto static structures may yield longer apparent cross-linking distances. Notably, longer distances were also observed for cysteine-to-cysteine cross-links in previous studies (Gutierrez, C., et al, *Anal Chem*, 2018; Jiao, F. et al, *Anal Chem*, 2023). In addition, non-satisfactory cross-links are often indicative of protein structural dynamics (Gutierrez, C., et al, *PNAS*, 2020; Leitner, A., et al, *PNAS*, 2014; Rampler, E., et al, *J. Proteome Res*, 2015). Taken together, we think it is reasonable not to apply strict distance restraints based solely on the nominal spacer arm length of SIA for this study. This part has been added to discussion.

Figure 5R. Spacer arm lengths of SIA and SBAP. SIA spacer arm should include 3 bonds (4.2

Å) instead of 1 bond (1.5 Å), whereas SBAP spacer arm should include 7 bonds (9.0 Å) instead of 5 bonds (6.2 Å).

Q3. *p. 8 & p.9: When the authors state the number of identified cross-links from BSA and HEK293, they must report in the main text at which FDR, on which level (CSM, XL, PPIs) and with which strategy (inter/intra separate, context-sensitive, etc.) the data was filtered. This information is missing in both manuscript and suppl. information, which strongly hinders reproducibility. The search engine ProteinProspector must be mentioned in the main text.*

A3: We apologize for the missing information. For BSA and in-cell XL-MS studies, we searched the XL-MS data against a concatenated target-decoy database composed of the entire Bos Taurus proteome (6,006 entries) and Homo Sapiens proteome (20,387 entries) for 293 cells. As a result, MSⁿ-based cross-link identification did not result in any decoy hits, indicating no false hits were found. This is not surprising as MSⁿ-based analysis is achieved through the integration of MS¹, MS² and MS³ data with linear peptide sequencing, yielding high-confidence identification. In comparison, MS²-based cross-link identification generally has higher FDR due to the limitations in identifying two cross-linked peptide sequences in a single spectrum and unavoidable database expansion (n^2). In this work, 1% FDR cut-off at the CSM level was applied for all MS²-based cross-link identification. We would like to clarify that Protein Prospector was mentioned in the method section. To address the reviewers' concerns, details on FDRs and search engine have been added in the main text as follows on page 9-10: "The MSⁿ-based analyses generated a total of 66,989 K-C cross-linked peptides from 1,612 proteins from HEK 293 cells with no decoy hits found (Supplementary Fig.6a, Supplementary Table 3). In addition, we performed MS²-based analysis for 2 SIA replicates, 3 SIAB replicates and 3 SBAP replicates (Supplementary Methods) to increase the depth of XL-MS data, identifying 57,059 K-C cross-linked peptides from 1,649 proteins with 1% FDR at the CSM level using Protein Prospector".

In addition, the FDR information was added to the Supplemental information in the section of "Identification of Cross-linked Peptides by stepped-energy HCD MS² analysis".

Q4. *p. 10: Max. 76% intra-link satisfaction on PDB structures is suspiciously low, especially with such relaxed distance cutoffs. Bartolec et al. PNAS 2023 have reported a > 89% satisfaction rate for intra-links (only DMTMM inter-chain links are below 80%). Also, the drastically low satisfaction of inter-links on PDB structures does not align with previous reports. On p. 16, line 345 resulting prediction match to 40% of cross-links which does not support the reliability of either the cross-link IDs or the AF prediction.*

A4: We thank the reviewers for the question. In current XL-MS studies, mapping of cross-links onto static structures is a widely used strategy to assess the plausibility of identified cross-links.

While distance satisfaction provides supportive evidence for cross-link validity, distance violations should not be interpreted as definitive evidence against it. While Bartolec et al. PNAS 2023 reported > 89% satisfaction rate for DSSO intra-links, lower distance satisfaction rates have also been reported for homobifunctional lysine cross-linkers by Wang, et al (*Nature Communications*, 2022,13:1468), in which only 55.21% and 78.99% intra-protein satisfaction rates were observed for a ten-standard protein mixture including BSA using DSS and a DOPA linker respectively. To gain further understanding on factors that may contribute to cross-link distance violations we have performed additional analyses as follows:

1. Protein dynamics. It is known that XL-MS analysis captures diverse and transient states in solution, revealing a conformational ensemble of protein structures, instead of one state as shown in static models. Abundant literature has indicated that not all cross-links must be satisfied in static structures due to protein dynamics and flexibility (Sinz, A. *Angew. Chem. Int. Ed.* 2018; O'Reilly, F.J. & Rappsilber, J. *Nat. Struct. Mol. Biol.* 2018; Gutierrez, C., et al, *PNAS*, 2020.). In addition, a molecular dynamics simulation analysis using homobifunctional reagents has revealed flexibility in cross-linkable distances that is not visible in static models (Merkley, E.D., et al, *Protein Sci.*, 2014). Specifically, SERBP1 interaction with ribosomal proteins, where we report 40% satisfaction for inter-protein cross-links, falls within this category of highly dynamic interactions that could not be observed using cryo-EM.

We have performed additional data analysis to further support the effect of dynamics on satisfaction rate, adding the following text and figures to the manuscript on page 25-26:

“To account for the influence of protein dynamics on the satisfaction rate, we examined whether cross-link distance satisfaction correlates with identification frequency across different linkers and biological replicates. We assume that cross-links observed in multiple replicates and linkers are more likely to originate from structurally stable regions. Indeed, cross-links identified by more than one linker also showed shorter median distances (Supplementary Fig. 9d). Likewise, cross-links detected more frequently across biological replicates exhibited lower median distances (Supplementary Fig. 9e–g) and higher satisfaction rates (Supplementary Fig. 9h), further supporting the notion that reproducibility is associated with structural stability.”

2. Cross-linker reactivity and rigidity. Our K-C linkers contain two reactive groups, i.e. NHS ester and haloacetamide, which have distinctly different reactivity towards their targeted residues. Compared to lysines, cysteines are much sparser and less exposed in protein structures. Moreover, cysteines can form disulfide bonds, and their reactivity is much more sensitive to their chemical microenvironments and redox states, which can impact protein conformations. To evaluate the chemical environments of cross-linked cysteines and lysines from in-cell XL-MS data, we plotted their pKa distribution (Supplementary Fig. 16). In comparison, cross-linked cysteines displayed a much wider range of pKa than cross-linked lysines, suggesting varied chemical environments for cysteines. Apart from the reactive groups, the spacer arm lengths from the MS-cleavage site to the targeted lysine and cysteine residues are different.

Taken together, the inherent asymmetry of lysine–cysteine cross-linking can lead to cross-links with unexpected 3-D orientation not reflected in static models, complicating the estimation of the maximum span between the cross-linked residues.

When comparing the three K-C cross-linkers, we find that there are also differences among them in the satisfaction rate. SIAB reached over 85% satisfaction for intra-protein cross-links while SIA and SBAP were at ~80% (Fig. 4a, Supplementary Fig. 9e-h). This can be explained by the relative rigidity of SIAB. In turn, SIA and SBAP cross-linking resulted in more cross-links identified (Supplementary Fig. 9b) due to their higher reactivity than SIAB.

Supplementary Fig. 16. Distribution of pKa values for cross-linked lysines (left) and cysteines (right) as calculated by PROPKA 3.1

Q5. p. 13, line 274: *Claiming that 59.1% of PPIs are new because they haven't been reported previously and don't match to any public repositories does not intrinsically indicate their novelty. This claim should be removed as those entries can also be interpreted as contradictory to prior studies, arising from potentially false-positive identifications.*

A5: We apologize for the confusion. Since XL-MS analysis identifies both PPI identity and residue-specific contacts, we meant that these experimentally derived interactions have not been reported by previous studies. To make the comparison clearer, we modified the PPI mapping approach, considering only inter-cross-linked PPIs for comparison. we revised the text as follows on page 16:

“In comparison, 13.8% (670) of the K-C XL-PPIs were found in published XL-MS studies, 11% (536) and 5.8% (283) were reported by the BioGrid and BioPlex databases, respectively, thus leaving 3,921 PPIs that have not been previously reported”.

Minor remarks:

Q1. *p. 3, line 31: What is an “enabling technology”?*

A1. We thank the reviewers for the question. To prevent any confusion, we changed “enabling technology” to “powerful technology” in the abstract.

Q2. *Cysteine residues are overall less abundant than lysine residues (~3% vs ~6%), making them a less attractive target for cross-linking. Can the K-C cross-linkers modify both oxidized C which form disulfide bridges and the reduced thiols? If one is limited to thiols, there are even less targets available.*

A2: We thank the reviewers for the question. K-C cross-linkers cannot modify oxidized cysteines involved in disulfide bonds. However, the cellular environment is normally in a reducing state and many cysteines exist in their free form. This enables K-C linkers to engage a subset of cysteines and effectively capture protein interactions that are not accessible via lysine-only chemistries. Although cysteines are less abundant and typically less solvent-exposed than lysines, this selective reactivity can be advantageous. K–C cross-linking would reduce nonspecific background, increase the likelihood of capturing biologically relevant interactions in cells and enable the detection of transient and/or low-abundance interactions that are inaccessible to lysine linkers. In our study, we demonstrate that K–C cross-links capture interaction and structural information complementary to that obtained using lysine–lysine cross-linkers (Supplementary Fig. 6, 8, 12). Given that the vast majority of *in vivo* XL-MS studies have relied solely on lysine-reactive cross-linkers, the incorporation of K–C cross-linking offers an opportunity to expand PPI coverages and enhance the structural resolution of cellular networks. We believe this work establishes a valuable foundation for employing heterobifunctional cross-linkers to advance systems structural biology approaches in the future.

Q3. *p. 9, line 168: Using stepped-HCD MS2, you identified 57059 K-C peptides, ~15% less compared to MS2-MS3 acquisition. This opposes other studies, e.g. <https://doi.org/10.1021/acs.analchem.1c04812> (Suppl. Fig. 1), <https://doi.org/10.1038/s41467-022-31701-w> (Fig.*

9), <https://doi.org/10.1021/acs.jproteome.8b00947> (Fig. 6). Can you explain this divergence?

A3: We thank the reviewers for their question. This study presents a comprehensive characterization of three K-C cross-linkers with multiple replicates associated with each linker. As our main goal is to establish the MSⁿ-based XL-MS analysis workflow for unambiguous identification of K-C cross-links, we put our major efforts on analyzing samples using LC MSⁿ. Since current MS²-based software is mainly designed for homobifunctional cross-linkers, it is difficult to systematically search and evaluate MS² data for K-C heterobifunctional cross-link identification. In contrast to MSⁿ-based analyses, it takes weeks to months to search and

finalize the search results for one set of proteome-wide data, making MS²-based analysis extremely time-consuming and challenging. Therefore, we performed MS² analysis on a smaller number of replicates per cross-linker compared to MS³-based analysis, with the primary aim of demonstrating the feasibility of MS²-based cross-link identification. In this study, we performed four replicates for SIA, six replicates for SIAB, and three replicates for SBAP using MS³-based analysis. In comparison, two replicates for SIA, three replicates for SIAB, and three replicates for SBAP were conducted using MS²-based analysis. Because the number of replicates differs between MS² and MS³ analyses, the resulting cross-linked peptide counts are not directly comparable. However, when we matched the number of MS³ replicates to the number of MS² replicates for comparison, we identified 36,679 K-C cross-linked peptides from MS³ analysis, which is approximately 35% fewer than the 57,059 K-C cross-linked peptides identified by MS² analysis. This reduction is consistent with findings from previous studies.

To clarify, we have modified our text as follows on page 10:

“In addition, we performed MS²-based analysis for 2 SIA replicates, 3 SIAB replicates and 3 SBAP replicates (Supplementary Methods) to increase the depth of XL-MS data, identifying 57,059 K-C cross-linked peptides from 1,649 proteins with 1% FDR at the CSM level using Protein Prospector”

Q4. Can you show that other search engines such as MSAnnika or XlinkX align with the data obtained from ProteinProspector?

A4: We thank the reviewers for the question. We have tried to search for the files by other searching engines such as MSAnnika and XlinkX. However, none of them worked for non-symmetric MS-cleavable cross-link identification. In addition, since MS-cleavability is introduced at the cysteine side chain after cross-linking and not within the linker region itself, it creates challenges for searching MS² XL data using existing software. Protein Prospector provides the flexibility to specify modified residues and applies mass modifications to identify cross-links, making it particularly suitable for analyzing our MS² data. However, this type of search is extremely time-consuming, limiting its broad use for large-scale analyses. Due to the overall scarcity of XL-MS datasets for heterobifunctional cross-linkers, we believe our work will serve as a valuable resource for the development of new tools to better identify these types of cross-links using MS²-based analysis.

Q5. p. 9, line 182: 847 proteins were found to be cross-linked only by K-C cross-linkers. Why is that? What are the unique features of those proteins? Is the number of Cs higher on those proteins compared to K-K cross-linked proteins?

A5: We thank the reviewers for the question. When examining the cysteine content in protein sequences, these 847 proteins appear to have a slightly higher median number of cysteines per

protein compared to those in the entire dataset and DSBSO (K–K) cross-linked proteins (8 vs. 7 vs. 7). Gene ontology analysis revealed that these 847 proteins are enriched more in mitochondrial components than the entire dataset (11.4% vs. 8.7%), consistent with the mitochondria's unique redox environment and central role in metabolic processes. However, in addition to the overall abundance of cysteine residues in proteins, several other factors critically influence whether a protein–protein interaction can be captured by a specific cross-linker. First, a cross-linkable residue pair—such as lysine and cysteine—must be in the desired spatial proximity and in an accessible conformation for the cross-linker to react. This depends not only on the protein structure but also on its dynamic conformational states *in vivo*. Second, the resulting cross-linked peptides are suited for MS detection and identification. This involves considerations such as peptide abundance, size, charge state, fragmentation behavior, and ionization efficiency. Therefore, the success of cross-link identification is determined by a combination of chemical accessibility, structural context, and analytical detectability—not solely by the frequency of reactive residues.

Q6. *p. 9, line 183: I don't understand this statement and corresponding Suppl. Fig. 5c, could you please rephrase it? Why can "K-C XL-proteomes contain proteins identified with lysine- and/or cysteine-containing peptides"? The "or" implies that it does not have to be K-C but also K-K or C-C when using those cross-linkers.*

A6: We apologize for the confusion. To clarify, we have rephased the sentence as follows on page 10-11:

“When protein interactions are captured using K–K cross-linkers, identification of all interacting proteins relies solely on cross-linked lysine-containing peptides. In contrast, K–C cross-linkers enable the identification of interaction partners through cross-linked lysine-containing peptides, cysteine-containing peptides, or a combination of both.”

Q7. *When you're comparing your K-C XL proteome to K-K XL proteomes you have reported earlier, you need to emphasize that DSBSO has a much higher cross-linkable distance than all 3 K-C cross-linkers. Please include a statement that this reduces fair comparability between the K-C and K-K datasets.*

A7: We thank the reviewers for the suggestion and have revised the main text on page 10: “Due to differences in both cross-linking chemistry and linker design—including reactive groups, target residue specificity and linker arm lengths—847 cross-linked proteins were uniquely identified in this study (Supplementary Fig. 6b) compared to our previously reported *in vivo* K-K XL-proteome revealed by DSBSO linker (14 Å)¹⁷. Unlike DSBSO, which targets lysine–lysine pairs, the three K-C linkers used here have shorter spacer arms and different reactivities, contributing to this increased proteome coverage.”

Q8. p. 12, line 236: *The relationship between AlphaFold quality metrics and cross-link satisfaction was reported in numerous works that should be cited. For example: <https://doi.org/10.15252/msb.202311544>, <https://doi.org/10.1101/2023.12.03.569778>, <https://doi.org/10.1101/2023.07.19.549678>*

A7: We thank the reviewers for the suggestion. We have revised the manuscript and cited these literatures.

Q9. p. 13, line 266: *As you're correctly stating that predicting protein fragments may improve satisfaction rates, why haven't you tried that for selected violating cross-links?*

A9: We thank the reviewers for the suggestion. Unfortunately, we were unable to determine an effective way to fragment the full-length proteins into smaller pieces. This requires new method developments for future studies.

Q10. p. 13, line 271: *You created a PPI network with 2007 nodes (corresponding to crosslinked proteins) and 6589 edges (PPIs?). Could you please change the terms nodes and edges to their biological meanings – also in Suppl. Fig. 8a? I could not find what 6589 refers to in the main text.*

A10: We thank the reviewers for the suggestion and have revised the text on page 13 “To further illustrate the human interactome, we constructed a K-C XL-PPI network containing 2,007 proteins and 6,589 interactions” and the annotations in supplementary Fig. 11a from “2,007 nodes, 6,589 edges” to “2,007 proteins, 6,589 interactions”.

Q11. *Suppl. Fig. 8b should be separated into several venn diagrams showing the overlap of the K-C data with other data or databases. Especially the overlap with the BioPlex data is extremely low in my experience.*

A11: We thank the reviewers for the suggestion and revised Suppl. Fig. 8 accordingly (now Supplementary Fig. 11). We changed the Figure into venn overlap format and separated the reported PPIs into Bioplex, Biogrid, and XL-references and revised the manuscript accordingly. We have also observed the lowest overlap between our XL-PPIs with the BioPlex database as mentioned by the reviewers.

Supplementary Fig. 11b. Comparison of the XL-PPIs from this study against BioGrid, BioPlex

databases and published XL-PPI data.

(*Published XL-PPIs data were derived from these studies: Liu, F., et.al., *Nat Commun*, 2018; Kuma, Y., et.al., *Mol & Cell Proteomics*, 2020; Wheat, A., et. al., *PNAS*, 2021; Jiao, F., et.al., *Anal Chem*, 2021; Jiao, F., et.al., *Anal Chem*, 2023; Tara. B., et.al., *PNAS*, 2023; Jiao, F., et.al., *J. Proteome Res*, 2024.)

Q12. *Suppl. Fig. 8c and p. 13, line 276 suggest that the majority of PPIs could not be mapped to the STRING network. Have you filtered the STRING database for certain evidences or did you consider all entries in the database? If you considered all entries, the overlap with STRING should be higher than ~10% (<https://doi.org/10.1038/s41592-024-02478-1>, Fig. 6b).*

A12: We thank the reviewer for the insightful comment and apologize for the confusion. There are two reasons. First, we filtered the XL-PPI data to include only high-confidence interactions determined by cross-linked peptides with unique sequences, thereby eliminating redundancies arising from cross-linked peptides with homologous sequences shared by multiple proteins. Second, we recently found that a portion of the STRING interaction data was accidentally lost during the conversion of accession numbers from Ensembl to SwissProt in our initial data export. To address this, we re-exported the full STRING database, including all available evidence types, for more comprehensive PPI mapping. As a result, we mapped a total of 559 high-confidence PPIs to STRING, with 70.7% of these interactions having STRING scores greater than 0.9. The main text has been revised accordingly on page 17.

Q13. *Fig. 3d needs fully labeled axis. Instead of using ProteomicsDB I suggest to search the RAW data again with MaxQuant and calculate an iBAQ specifically for the samples used in this study. A plot as in <https://doi.org/10.1021/acs.analchem.3c04682> Fig. 2 would strengthen the claim that cross-links were identified throughout the whole proteome.*

A13: We thank the reviewers for the suggestion. In this work, our data acquisition was focused on cross-linked peptides, and only peptides with charge 4+ and up were selected for MS² and MSⁿ analyses. Unlike linear peptides, which directly reflect protein abundance and are commonly used to calculate iBAQ values in shotgun proteomics, the abundance of cross-linked peptides does not directly correlate with protein abundance. Instead, it is more influenced by the availability, accessibility, spatial proximity, and detectability of cross-linkable residue pairs at protein interaction interfaces. Additionally, when a cross-link forms between two proteins, their cellular abundances are often different, further decoupling cross-linked peptide abundance from individual protein levels. At the moment, there is no existing software available that can estimate protein abundance based on cross-linked peptides alone. Thus, protein abundance obtained from the ProteomicsDB would be better suited for evaluating the content of the XL-proteome. As suggested, we have added the x and y axis in Figure 3d.

Figure 3d. Abundance distribution of the 1,856 proteins found in the K-C XL-proteome that have iBAQ values in ProteomicsDB.

Q14. p. 17, line 356: 12% of loops in eEF1G is objectively not a “high percentage”, please rephrase.

A14: We thank the reviewer for the suggestion. We have revised the manuscript from “Structure determination of this complex is challenging due to a high percentage of long disordered regions connecting the globular domains” to “Structure determination of this complex is challenging due to the long-disordered regions connecting the globular domains” on page 20.

Q15. p. 19, line 400: You mention more than once the benefit of Cys oxidation induced cleavability for chemical proteomics. Please include in the discussion a more detailed hypothesis of what can be gained from applying this method?

A15: We thank the reviewer for the suggestion. The following text has been added in discussion on page 22-23:

“In chemical proteomics, haloacetamides are commonly used for global cysteine reactivity profiling to determine hyper-reactive cysteines (Xiao, H., et al., *Cell*, 2020; Kuljanin, M., et al, *Nature biotechnology*, 2021) and studying cysteine oxidation to elucidate cellular redox states (Weerapana, E., et. al., *Nature*, 2010; Fu, L., et al., *Nature Protocols*, 2020). Since the oxygen atoms incorporated during cysteine oxidation by chloramine-T are derived from water, an isotope label can be introduced during this process to enable quantitative analysis. Specifically, cysteine-labeled peptides could be oxidized in light (H_2O^{16}) and heavy (H_2O^{18}) water to generate light and heavy labels, allowing for quantitative comparison of cysteine reactivity or oxidation states without the need for synthesizing isotopically labeled reagents. In addition, due to the low abundance of cysteines, alkyne-tagged haloacetamides are often used to allow click-chemistry-based enrichment by conjugating an affinity tag such as biotin (Fu, L., et al.,

Nature Protocols, 2020; Fu, L., et al., *Nature Chemical Biology*, 2023). Although such enrichment enhances the detectability of cysteine-containing peptides, the attachment of a large chemical moiety on cysteines after click conjugation could complicate their MS identification. However, by incorporating an oxidation-induced cleavability, the covalently attached group can be selectively removed during MS/MS analysis, simplifying peptide analysis and facilitating their accurate identification. Moreover, similar strategies can be applied to facilitate the discovery of haloacetamide-based covalent drugs (Liu, Q., et al., *Chemistry & biology*, 2013; Resnick, E., et al., *Journal of the American Chemical Society*, 2019). Therefore, the established strategy would benefit chemical proteomics by facilitating cysteine identification and characterization, broadening the impact of the work presented here.”

Q16. *p. 20, line 432: You argue that SIAB cross-links are more in agreement with the distance threshold due to its lesser rigidity. Could you elaborate whether the rigidity of the cross-linker still plays a role when it is bound to a flexible lysine? Moreover, SIA is 7x shorter than SIAB, which intuitively should have the highest effect on whether a cross-link stays within a 30 Å cutoff.*

A16: We thank the reviewer for the insightful comments. Among the three selected linkers, SIAB appears to be the most rigid. However, apart from rigidity, we believe the observed differences could be influenced by other factors. As detailed in the response to the reviewers’ major point Q2 (see **Figure 5R**), the spacer arm lengths for SIA and SBAP were determined by Spartan as 4.2 Å and 9.0 Å, respectively, while SIAB’s spacer arm remained as 10.6 Å. In general, it is anticipated that the shortest linker SIA would form cross-links with the shortest distances. However, our data has shown that SIA was able to catch cross-links with distances similar to those of SIAB and SBAP cross-links. Among the three linkers, SIA and SBAP have much better solubility in aqueous buffer than SIAB. Based on peptide labeling experiments, SIAB exhibited the slowest cross-linking kinetics. The slower reaction rate makes SIAB more suited for capturing stable interactions that are better preserved in static structures, resulting in a better agreement with the expected structural distance threshold. In contrast, the faster-reacting SIA may capture more flexible or transient interactions. XL-MS analysis of the two selected standard proteins suggests that SIA can capture cross-links with much longer distances than expected. As mentioned above, we suspect that this observation is most likely attributed to the chemical properties and high reactivity of SIA, as well as cysteine’s reactivity and dynamic behavior that rely heavily on its microenvironment. Taken together, we think it is reasonable not to apply strict distance restraints based solely on the nominal spacer arm length of SIA for this study. This information has been implemented in the main text and discussion.

Q17. *On the same line: According to Suppl. Fig. 11, K-K cross-links are more prone to fall into loop regions, potentially yielding more overlength links when mapped on predicted structures*

as AF might fail to position those regions correctly. K-C linkages fall predominantly into Helices and Sheets, which usually have higher pLDDT scores. Would you not think that this should be reflected in higher cross-link satisfaction rates

A17: Thank you for this thoughtful observation. Indeed, our data in Supplementary Fig. 11 show that K-K cross-links are more frequently located in loop regions, which are generally more flexible. In contrast, K-C cross-links are enriched in helices and sheets, which tend to be more structurally constrained. However, while K-K cross-links do show a slightly higher satisfaction rate overall, the difference is not as pronounced as might be expected based solely on secondary structure. This may be due to a combination of factors, including conformational variability, cross-linker rigidity, and reactivity as we discuss in response to Q4 and Q16.

Q18. *p. 22, line 474: Was the mixture “reacted with each linker...in dark” because the cross-linkers are light sensitive? In the ThermoFisher User Guide for SIA (https://assets.thermofisher.com/TFS-Assets/LSG/manuals/MAN0011372_SIA_UG.pdf) it is not mentioned to incubate the linker in the dark.*

A18: We thank the reviewer for the question. The reaction was performed in the dark because the halogen acetamide compounds used are not stable when exposed to light. To prevent degradation and ensure reagent stability, all reactions were carried out under dark, which is the same as commonly used cysteine alkylation conditions (Salvatore, S. & Chait, B.T. *Anal. Chem.* 1998).

Q19. *p. 24, line 519: “Structures were filtered out if numbering of the aa did not match the sequence.” Does that mean that no structure mapping was performed the numbering of the PDB was off by even 1 position from the numbering of your cross-linking result? This would remove a lot of PDB structures from the pool of accepted structures because the numbering of PDB structures is seldomly in accordance with the numbering of a FASTA retrieved from UniProt. The authors should find means to consider mapping links to all PDB structures and fix the numbering by calculating a protein-specific offset if needed.*

A19: We thank the reviewer for this important point. In our current analysis, we excluded PDB structures where the residue numbering did not exactly match the UniProt sequence, even if the offset was just one position. However, each protein was typically mapped to multiple structural models, and in practice, over 83% of cross-links with structural data had at least one structure retained after residue mapping. Therefore, the number of cross-links lost due to strict residue matching was relatively small. We acknowledge the reviewer’s suggestion to implement a protein-specific offset or alignment-based correction for residue mismatches. This is a valuable recommendation, and we agree that incorporating such an approach in future analyses could improve structural coverage and reduce data loss due to minor numbering discrepancies.

Q20. *Fig. 4d needs a y-axis label. “AlphaFold confidence” can be said in the title, but the y-axis should contain the actual metric (PAE).*

A20: We thank the reviewer for the suggestion. We have revised the Figure 4d, adding “PAE” as its y-axis.

Q21. *Fig. 6: add a density plot visualizing cross-link satisfaction.*

A21: We thank the reviewer for the suggestion. We have added the density plot to visualizing the cross-link satisfaction (Figure 6c).

Q22. *The RAW files uploaded to PRIDE may be of general interest as the data availability for heterobifunctional cross-linkers is sparse, thus it might be useful to upload an assignment table for RAW files to experiment and/or FASTA file and/or figure they were used for.*

A22: We thank the reviewer for the suggestion. We have uploaded the assignment table for raw files and the FASTA files.

Q23. *The XL-MS community puts a strong focus on uniform data sharing, so please consider uploading your data to PRIDE in the newest mzIdentML format.*

A23: We thank the reviewer for the suggestion. We have uploaded the mzIdentML files to the PRIDE.

Q24. *The title could emphasize more the heterobifunctional character of the cleavable cross-linker.*

A24: We thank the reviewer for the suggestion. However, the cysteine-enabled cleavability is not limited to heterobifunctional cross-linkers. Therefore, we feel the current title makes it more general for a broad application in the future.

Q25. *For density plots, consider showing the median, too.*

A25: We thank the reviewer for the suggestion. We added the median annotation to the density plot in Figure 4b,c, Figure 5a-d, Figure 6c, Figure S9d,i and Figure S14b.

Detailed responses to reviewers' comments:

Note: To distinguish the question numbers from current and previous revisions, we marked questions for this revision as QR#. Figures included solely to address reviewers' comments are labeled as **Figure #R** (# represents figure number for responses). Figures that appear in the manuscript retain the same figure number and description here as in the main text.

Reviewer #4

***QR1.** Major Concern A3: The authors state that their MSⁿ-based strategy yielded no decoy identifications and thus no FDR estimation, which is highly unusual. Their search was conducted against the entire *Bos taurus* (6,006 entries) and *Homo sapiens* (20,387 entries) proteomes. The MSⁿ-based strategy searches the two linked peptides individually using the precursors obtained in MS2 and fragment ions in subsequent MS3. To the best of my knowledge, all peptide searches generate decoy hits and FDR is calculated based on number of target and decoy identifications. Could it be that ProteinProspector simply does not export them so the authors do not see any decoy hits? Regardless, FDR estimation (even for MS²–MS³) is essential for validation, and this must be performed and reported; without it, the results remain questionable.*

AR1: We thank the reviewer for this question and apologize for any confusion. In this work, to identify cross-linked peptides from LC MSⁿ data, our data analysis consists of two major steps: 1) Protein Prospector search of MS3 data to identify linear peptides carrying defined cross-link fragments; 2) integration of MS1, MS2 and MS3 data to identify cross-linked peptides using xl-Tools. For a cross-link to be successfully identified, two linear peptide constituents must be identified from the MS3 within the same MS2 scan, carry unique cross-linker modifications due to characteristic fragmentation, and satisfy a defined mass relationship to their parent ion measured in MS1. The low FDRs in our MSⁿ-based cross-link identifications likely reflect the stringent requirements of MSⁿ data integration, enabling high-confidence and unambiguous cross-link assignments. This MSⁿ-based XL-MS data analysis workflow follows the same procedures as those used for our previously developed sulfoxide-containing MS-cleavable cross-linkers (Kao, A., et al, *Mol Cell Proteomics*, 2011; Yu & Huang, *Anal Chem*, 2018). The requirements for MSⁿ integration have been added in Supplementary Methods.

To address the reviewer's concern, we re-examined our search results carefully and performed several additional analyses. To properly assess the search results, we evaluated the FDRs for both MS3-based linear peptide search results and MSⁿ based cross-link identification, which were calculated using a target-decoy strategy as commonly used in proteomics studies.

First, to illustrate the accuracy of our cross-link identification, we used a SIA cross-linked standard peptide (i.e. LR9 (Ac-LADVCAHER) cross-linked to SR8 (Ac-SAKAYEHR)) as an example and searched its MS3 data in Protein Prospector against a target-decoy database containing these two peptide sequences and the entire yeast database (6,727 entries) concatenated with their random sequences. As a result, 725 PSMs (linear peptide spectrum matches) were identified, including 4 yeast and 4 decoy peptide sequences, corresponding to a 1.1% FDR. After MSⁿ data integration using XL-tools, 21 CSMs (cross-link spectrum matches) were identified, all corresponding to the expected SIA cross-linked LR9-SR8 peptides. No decoy or yeast sequences were observed, resulting in an FDR of 0% at the CSM level. The same analyses were performed on LC MSⁿ data for SIAB and SBAP cross-linked standard peptides (LR9 cross-linked to SR8), yielding similar results; only the expected cross-linked peptides were identified, with no decoy matches and thus an FDR of 0% at the CSM level for these standard peptide samples. These results indicate that MSⁿ data integration effectively eliminates false peptide sequences, leading to low FDRs at the cross-link identification level.

Next, we re-analyzed the BSA data from one replicate for all three linkers, SIA, SIAB, and SBAP. To better assess FDRs at the CSM level, we examined their relationship to PSM-level FDRs generated in Protein Prospector at the MS3 level. For this analysis, linear peptide reports were obtained in Protein Prospector at three PSM-level FDR thresholds (1%, 5%, 10%), exported, and then used for MSⁿ data integration in xl-Tools. As shown in Figure R1, decoys were not identified at the CSM level when PSMs were exported with FDR ≤ 5%; but were detected with 10% PSM-level FDR. These results indicated that CSM-level FDRs increase proportionally with PSM-level FDRs across all linkers as expected.

Figure R1. Evaluation of CSM-level vs. PSM-level FDRs for cross-linked BSA across the three linkers.

For the BSA datasets described in the manuscript, the PSM-level FDRs ranged from 1.50% to 2.34% for SIA, 0.75% to 1.50% for SIAB, and 0.68% to 1.73% for SBAP across biological

replicates, all remaining below the 5% threshold and yielding a CSM-level FDR of 0%. For the ALDOA datasets, the PSM-level FDRs ranged from 1.05 to 1.14% for SIA, 0.41 to 0.45% for SIAB, and 0.79 to 0.86% for SBAP across biological replicates, with a CSM-level FDR of 0% also observed.

We agree that it is unusual to have proteome-wide cross-link identifications without any decoy hits. Upon re-examining the data, we discovered a minor reporting error that resulted in the omission of decoy-containing CSMs. We thank the reviewer for bringing this issue to our attention, which enabled us to identify and resolve it. After resolving the issue, we recalculated the FDRs for our *in vivo* XL-MS data at the CSM level, resulting in average FDRs of 0.07%, 0.1%, and 0.03% for SIA, SIAB, and SBAP, respectively, across biological replicates. We have added this information to Supplementary Fig. 6, introducing a new panel d as shown below.

Supplementary Fig. 6d. CSM-level FDRs for MSⁿ-based identification of SIA, SIAB, and SBAP cross-linked peptides across biological replicates. SIA: 4 replicates; SIAB: 6 replicates; and SBAP: 3 replicates.

These changes have been updated in the main text on page 10: “The MSⁿ-based analyses generated a total of 66,989 K-C cross-linked peptides from 1,612 proteins in HEK 293 cells, with average FDRs of 0.07%, 0.1%, and 0.03% at the CSM (cross-link spectrum match) level for SIA, SIAB, and SBAP, respectively, across biological replicates (Supplementary Fig. 6a, d, Supplementary Table 3)”.

QR2: *This concern also relates to Minor A4, where I suggested cross-validation using alternative software (e.g., MSAnnika, XlinkX), given that ProteinProspector is not widely adopted in this field. While the authors note that MSAnnika and XlinkX do not support asymmetric crosslinkers, xiSearch offers better flexibility. Could the search be done with xiSearch to strengthen the findings?*

AR2: We thank the reviewer for this suggestion. For clarification, we would like to emphasize that most currently available cross-link identification software support asymmetric cross-linkers to some degree. However, due to the limited availability of XL-MS studies employing asymmetric heterobifunctional MS-cleavable cross-linkers, existing search algorithms are not fully optimized to consider the expected fragmentation patterns for cross-linkers such as SIA, SIAB, and SBAP. While xiSearch indeed provides flexibility for diverse cross-linkers, one of the main downsides of its cross-linker configuration is that it does not allow association of specific cross-link fragments with corresponding cross-linked sites. Given that SIA, SIAB, and SBAP are heterobifunctional with single asymmetric cleavage, one can expect that lysine and cysteine residues will always carry a single type of cross-linker fragment moiety. However, the inability to specify which fragment (short or long arm) is associated with each residue inflates the search space for each cross-link, decreasing the accuracy of the identification by reducing the percentage of matched fragment ions. To circumvent this, we conducted additional analyses using another widely adopted tool, MeroX, which does allow these cross-linkers to be configured properly. As an example, we used LC-MS/MS data from SIA cross-linked BSA and conducted the same search with MeroX against a database containing BSA and 26 proteasome proteins as decoys, following the procedure used for Protein Prospector. Comparison of the same scans identified by both tools revealed essentially identical cross-linked peptides, supporting the reliability of our identifications. At a 1% FDR cutoff, MeroX reported 103 unique BSA cross-linked peptides, including two BSA-proteasome linkages, whereas Protein Prospector identified 160 unique BSA cross-linked peptides with no proteasome linkages. Notably, the majority of MeroX identifications (86 of 101, 85.1%) overlapped with Protein Prospector, demonstrating strong consistency between the two platforms (Figure 2R). As shown, cross-linked peptides uniquely identified by MeroX were largely lower-score matches. In contrast, the score distribution of cross-linked peptides identified by both software and those uniquely to Protein Prospector was more similar, indicating comparable confidence in Protein Prospector identifications of both shared and unique cross-links relative to MeroX. Collectively, these results confirm the robustness of our findings and highlight the value of this study as a resource for developing future software tools to identify cross-linked peptides generated by asymmetric heterobifunctional MS-cleavable cross-linkers.

Figure R2. Comparison of cross-linked peptides identified from SIA cross-linked BSA by Protein Prospector and MeroX. a Overlap of cross-linked peptides identified by Protein Prospector and MeroX at a 1% FDR cutoff. b MeroX score distribution of cross-linked peptides identified by both tools (green) and by MeroX only (pink). c Protein Prospector score distribution of cross-linked peptides identified by both tools (green) and by Protein Prospector only (blue).

QR3: Minor Comment A18: Please include a brief explanation in the Methods section to highlight the reagent's instability, ensuring readers are aware of this limitation.

AR3: We thank the reviewer for the suggestion. We have added this information in the Methods part on page 28: "Due to their light sensitivity, these chemicals were stored and handled in the dark during the cross-linking reactions."

Reviewer #5 (Substitute Reviewer #1)

Comment #1: Question #1, Answer to Question #1.

QR1. The authors have improved the manuscript by clearly indicating their replicates. However, they must improve Table S6, with all data regarding the exact cross-links covering those complexes. It is extremely challenging now to search and map those on protein complexes for this reviewer, as well as seeing how many, or which cross-links were captured per reported CORUM complex.

AR1: We thank the reviewer for the question and apologize for any confusion. Our *in vivo* XL-MS analysis focused on global interaction profiling, which by itself cannot define protein complex compositions. Therefore, we relied on established protein complex definitions from the CORUM database (<https://mips.helmholtz-muenchen.de/corum/>). For our analysis, we

uploaded the list of cross-linked proteins identified in this study to CORUM and determined how many of its annotated complexes were represented in our dataset. This has been clarified in the method section. As this analysis does not require linkage information, the original Table S6 only included the information on the list of CORUM complexes that were present in our dataset, along with percentage of subunit recovery of the complex and the number of subunits detected. Because cross-link data involve complex inter- and intra-protein interactions and many proteins participate in multiple complexes, it is not straightforward or feasible to directly integrate cross-link information into this table.

However, we agree with the reviewer that including cross-link information within each complex would be valuable. To address this, we mapped the identified K-C linkages to protein complexes that have high-resolution structures in the PDB database. In total, 5,904 K-C linkages were mapped across 573 PDB complexes. This detailed linkage information is now provided in a new supplemental Table S6B, while the original CORUM complex analysis table has been renumbered as Table S6A. These updates have also been incorporated in the main text on page 10:

“In addition, 5,904 K-C linkages were mapped across 573 protein complexes that have high resolution structures in the PDB database (Supplementary Table 6B).”

Comment #3: Question #3, Answer to Question #3:

QR2: Properly addressed, I guess it is now Figure S9. However: The authors must understand and revise the presentation of AlphaFold models in the text, versus PDB models: AlphaFold models are predictions, whereas PDB models are experimental structures. Cross-links cannot be validated by a prediction, but a prediction is validated by the cross-linking data.

AR2: We thank the reviewer for the suggestion and have revised the presentation of AlphaFold models in the text accordingly.

Comment #6: Question #6, Answer to Question #6.

QR3: The authors did not respond to this comment. They must clarify if they benchmarked cross-linking time or not.

AR3: We thank the reviewer for the question and apologize for any confusion. In our

experiment, we performed oxidation of K-C cross-linked peptides to enable the MS-cleavability of cross-linked cysteines. The previous reviewer asked if the oxidation time was evaluated. As stated in our previous response, we compared 1-minute and 5-minute oxidation treatments and found no detectable differences in the results. To clarify, we did not benchmark the cross-linking time. The final *in vivo* cross-linking condition was based on our standard peptide tests, our prior experience with in-cell cross-linking (e.g. Wheat, A., et al, PNAS, 2021) and initial LC MSⁿ results.

Detailed responses to reviewers' comments:

Reviewer #4

***QR1.** While I still had some reservations about the FDR control method in the ProteinProspector pipeline, in this round of the revision the authors compared their results with an independent search engine, MeroX. The two pipelines showed strong agreement (85% overlap) in identification. MeroX reports an entrapment-based false-positive rate of only ~2% at a 1% FDR, while ProteinProspector identified zero false positives at any FDR threshold. I think this information should be included in the manuscript, which also emphasizes their pipeline is somewhat unique and robust. Then I would be happy to accept its publication.*

AR1: We thank the reviewer for the suggestion and have added this information in the manuscript discussion section on page 22.

“Notably, because few XL-MS studies have employed asymmetric heterobifunctional MS-cleavable cross-linkers, most existing search algorithms are not yet fully optimized to account for their characteristic fragmentation patterns, such as those produced by SIA, SIAB, and SBAP. In contrast, Protein Prospector enables linker-specific modification settings and has proven effective in identifying K-C cross-links from both MS2 and MS3 data. Collectively, this work represents a technological advancement in probing cellular networks and establishes a valuable resource for developing future software tools to improve identification of cross-linked peptides generated by asymmetric heterobifunctional MS-cleavable cross-linkers.”